# Genomic profile of advanced breast cancer in circulating tumour DNA

Belinda Kingston [1], Rosalind J. Cutts[1], Hannah Bye[2], Matthew Beaney[1], Giselle Walsh-Crestani[1], Sarah Hrebien[1], Claire Swift[1], Lucy S. Kilburn [3], Sarah Kernaghan[3], Laura Moretti [3], Katie Wilkinson[3], Andrew M. Wardley[4], Iain R. Macpherson [5], Richard D. Baird [6], Rebecca Roylance[7], Jorge S. Reis-Filho [8], Michael Hubank[2], Iris Faull [9], Kimberly C. Banks [9], Richard B. Lanman [9], Isaac Garcia-Murillas [1], Judith M. Bliss[3], Alistair Ring [10]✉ & Nicholas C. Turner[1,10]✉

The genomics of advanced breast cancer (ABC) has been described through tumour tissue biopsy sequencing, although these approaches are limited by geographical and temporal heterogeneity. Here we use plasma circulating tumour DNA sequencing to interrogate the genomic profile of ABC in 800 patients in the plasmaMATCH trial. We demonstrate diverse subclonal resistance mutations, including enrichment of *HER2* mutations in HER2 positive disease, co-occurring *ESR1* and MAP kinase pathway mutations in HR + HER2− disease that associate with poor overall survival ($p = 0.0092$), and multiple *PIK3CA* mutations in HR + disease that associate with short progression free survival on fulvestrant ($p = 0.0036$). The fraction of cancer with a mutation, the clonal dominance of a mutation, varied between genes, and within hotspot mutations of *ESR1* and *PIK3CA*. In ER-positive breast cancer subclonal mutations were enriched in an APOBEC mutational signature, with second hit *PIK3CA* mutations acquired subclonally and at sites characteristic of APOBEC mutagenesis. This study utilises circulating tumour DNA analysis in a large clinical trial to demonstrate the subclonal diversification of pre-treated advanced breast cancer, identifying distinct mutational processes in advanced ER-positive breast cancer, and novel therapeutic opportunities.

[1] The Breast Cancer Now Toby Robins Research Centre, The Institute of Cancer Research, London, UK. [2] Centre for Molecular Pathology, Royal Marsden Hospital, London, UK. [3] ICR-CTSU, The Institute of Cancer Research, London, UK. [4] NIHR Manchester Clinical Research Facility at The Christie, Manchester Academic Health Science Centre & Division of Cancer Sciences, School of Medical Sciences, Faculty of Biology Medicine & Health, University of Manchester, Manchester, UK. [5] The Beatson West of Scotland Cancer Centre, Glasgow, UK. [6] Cancer Research UK Cambridge Centre, Cambridge, UK. [7] University College London Hospitals NHS Foundation Trust, London, UK. [8] Memorial Sloan Kettering Cancer Centre, New York, NY, USA. [9] Guardant Health, Inc., Redwood City, CA, USA. [10] Ralph Lauren Centre for Breast Cancer Research, Royal Marsden Hospital, London, UK. ✉email: Alistair.ring@rmh.nhs.uk; nicholas.turner@icr.ac.uk

Cancers evolve over time[1], and a metastatic cancer can harbour a different mutational profile following relapse compared to the primary tumour[2–8], in part due to genetic events that can be selected by therapy as mechanisms of resistance[9,10]. Traditionally tissue biopsies have been used to characterise metastatic breast cancer, and large scale genomic sequencing efforts have made remarkable progress in defining the genomic landscape of metastatic breast cancer through tumour tissue sequencing[11–15]. However, diagnostic tissue biopsies provide a genomic snapshot limited to the primary tumour, and repeat advanced disease biopsies are limited by sampling bias, in particular in the presence of spatial heterogeneity, where individual metastases may develop different resistance mechanisms[16–18], and are invasive[19]. Circulating tumour DNA (ctDNA), released into the systemic circulation following tumour cell death[20,21], is theoretically an admixture of tumour DNA from heterogeneous metastatic sites, and may more fully describe tumour heterogeneity[22].

In metastatic sequencing studies of advanced breast cancer, the most major differences to primary breast cancer have been identified in hormone receptor (HR+) positive breast cancer[12]. In HR+ breast cancer oestrogen receptor mutations (ESR1)[23], MAP kinase (MAPK) pathway mutations[11,24], and transcription factor alterations[11] such as ARID1A mutations[11,25,26], are acquired as mechanism of resistance to prior endocrine therapy. In single site biopsies, these routes to endocrine therapy resistance are mutually exclusive[11]. Tumour autopsy studies have demonstrated the substantial prevalence of geographical heterogeneity, where individual metastases may have different genomic profiles[27], raising uncertainty over whether genomics studies based on single site biopsies have captured the full picture of advanced breast cancer genomics. In triple negative and HER2 positive (HER2+) breast cancer, no major changes have been identified in the genomics of advanced breast cancer, compared with primary breast cancer[11,12].

Here we define the genomic profile of metastatic breast cancer using ctDNA sequencing from patients within plasmaMATCH, a prospective platform trial leveraging ctDNA analysis in patients with metastatic breast cancer for which the primary outcomes have been published[28]. In this ad-hoc analysis we investigate how the profile of somatic genetic alterations in ctDNA differs from that obtained by tumour tissue sequencing. Using the rich clinical data set associated with the clinical trial, we explore the clinical and pathological associations of advanced breast cancer genomics, and define the processes that generate the genomic diversity of metastatic breast cancer.

## Results

**PlasmaMATCH circulating tumour DNA sequencing.** Between 21st December 2016 and 26th April 2019, 1,051 patients were enroled into plasmaMATCH (Supplementary Fig. 1a). Baseline pre-treatment ctDNA targeted sequencing results were available for 800 patients[28] (available in Supplementary Data 1). Of the patients with available targeted sequencing, 64.4% (N = 515) had hormone receptor positive (HR+) HER2 negative (HER2-, lack of HER2 over-expression and/or gene amplification) disease, 9.1% (N = 72) were HER2 positive, and 17.3% (N = 138) had triple-negative breast cancer (TNBC, Supplementary Table 1).

Plasma samples were sequenced by duplex error corrected sequencing with a clinical diagnostic panel targeting 74 cancer genes (Supplementary Fig. 1b). Overall, 92.9% of patients were found to have at least one ctDNA alteration. The most frequently altered genes were TP53 (44.1%), PIK3CA (34.9%), ESR1 (33.1%), GATA3 (11.0%), ARID1A (7.8%) and PTEN (6.9%, Fig. 1a). The frequency of TP53, PIK3CA, ESR1 and GATA3 mutations varied according to breast cancer subtype (Fig. 1b) as previously

described[11,12,29]. We also identified novel subtype associations, with HER2 (also known as ERBB2) mutations enriched in HER2+ disease compared to HR+HER2- (q = 0.05) and TNBC (q = 0.005, Fig. 1b). HR+HER2- disease had significantly more pathogenic alterations per patient than HER2+ disease (3.0 vs 2.2, p = 0.03) and TNBC (3.0 vs 1.8, p < 0.0001, Fig. 1c).

Copy number (CN) alterations in ctDNA were also associated with breast cancer subtype (Fig. 1d and Supplementary Fig. 2), with CN alterations in MYC, PIK3CA, EGFR, CCNE1, CDK6, BRAF and MET more common in TNBC (q = 0.01, q < 0.0001, q = 0.001, q < 0.0001, q = 0.0003, q = 0.0002 and q = 0.01, respectively, Fig. 1d). FGFR1 and CCND1 CN alterations were more common in HR+HER2- breast cancer (q = 0.01 and q < 0.001, respectively). A small subset (1.7%) of HR+HER2- and TNBC, assessed in prior tissue, had HER2 amplification detected in ctDNA (Fig. 1d), likely reflecting acquisition of HER2 amplification.

We assessed the sensitivity of targeted sequencing to identify droplet digital PCR (ddPCR) mutation calls in targetable hotspots within PIK3CA, HER2, AKT1 and ESR1. Within the 682 patients who underwent ctDNA testing with both technologies, the targeted sequencing demonstrated a high sensitivity of 90.9% in identifying mutations. For mutations with ddPCR allele frequency <1%, the targeted panel sensitivity was 80.9% (Supplementary Fig. 3).

**Polyclonal genomic resistance in ctDNA.** Mutated genes showed tendency for patterns of co-enrichment and mutual exclusivity. PIK3CA and AKT1 alterations were mutually exclusive (q = 0.001)[11,29], and ESR1 and TP53 alterations were mutually exclusive (q < 0.0001, Fig. 2a)[12,30,31]. In agreement with metastatic tissue sequencing datasets[11], NF1 and TP53, and PIK3CA and HER2 tended to co-occur (p = 0.05 and p = 0.02, respectively, Fig. 2a). Similar patterns were noted in HR+HER2- disease where PIK3CA and GATA3 (p = 0.03) and ESR1 and TP53 (p = 0.002) tend to be mutually exclusive, suggesting distinct mechanisms of endocrine resistance in TP53 mutant cancer. PIK3CA and KRAS alterations (p = 0.02), and PTEN and NF1 (p = 0.02) alterations were found to be co-enriched (Supplementary Fig. 4)

In metastatic tissue sequencing datasets, alterations within the MAPK pathway and ESR1 mutations are mutually exclusive in HR+HER2- disease[11]. In contrast, in ctDNA sequencing MAPK alterations were more common in patients with ESR1 mutations overall (p = 0.001, Fig. 2b), and in HR+HER2- breast cancer (p = 0.02, Fig. 2b). Allele fractions of ESR1 and MAPK alterations in the same patient did not overlap, indicating these mutations existed in different clones (Supplementary Fig. 5). Furthermore, MAPK alterations were substantially enriched in patients with polyclonal ESR1 mutations as compared to single ESR1 mutations (39.3% versus 19.6% respectively, p = 0.0004, Fig. 2b), identifying a subset of oestrogen receptor (ER) positive breast cancers that develop polyclonal genomic resistance (Fig. 2c). Overall survival in patients with HR+HER2- disease and alterations in both ESR1 and MAPK pathway was 7.9 months and in patients wild-type for both ESR1 and MAPK pathway was 18.5 months (Fig. 2d), with polyclonal resistance associating with poor survival.

**Circulating versus tissue based genomic profiling.** We investigated how the profile of somatic genetic alterations in breast cancer ctDNA compared to that obtained by tissue sequencing, in a large metastatic breast cancer tissue sequencing dataset (MSK-IMPACT™)[11] (Supplementary Tab. 2). In HR+HER2- disease, we identified enrichment for pathogenic alterations in ESR1 and

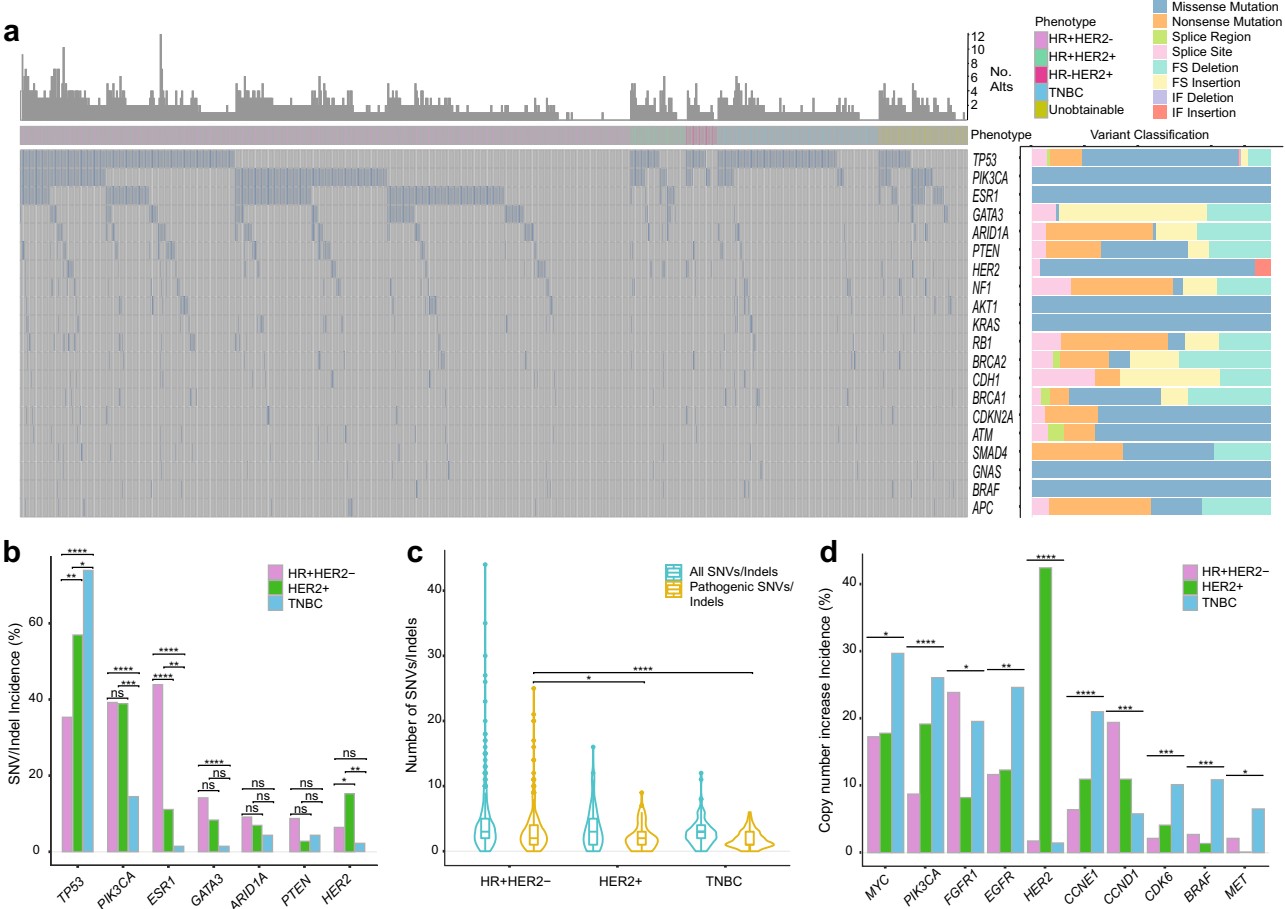

**Fig. 1 Mutation profile of advanced breast cancer determined by ctDNA sequencing. a** Mutational profile of advanced breast cancer determined by ctDNA targeted sequencing of 800 patients in the plasmaMATCH trial. Displayed are mutations and indels likely to be pathogenic ("Methods"), summarised by gene for each patient. Top bar refers to total counts of pathogenic mutations per patient. *Right*, variant classification of the alterations within each gene. FS, frameshift; IF, in-frame. **b** Breast cancer subtype association of the most frequently mutated genes within patients with known phenotype (HR + HER2- N = 515, HER2+ N = 72, TNBC N = 138). Comparison with false discovery corrected two-sided Fisher's exact tests (*TP53*: HR + HER2- vs HER2+ q = 0.003, HR + HER2- vs TNBC q < 0.0001, HER2 + vs TNBC q = 0.04; *PIK3CA*: HR + HER2- vs TNBC q < 0.0001, HER2 + vs TNBC q = 0.0008; *ESR1*: HR + HER2- vs HER2 + q < 0.0001, HR + HER2- vs TNBC q < 0.0001, HER2 + vs TNBC q = 0.008; *GATA3*: HR + HER2- vs TNBC q < 0.0001; *HER2*: HR + HER2- vs HER2 + q = 0.05, HER2 + vs TNBC q = 0.005). ns, not significant. **c** Patient mutation frequency split by breast cancer subtype, overall and likely pathogenic mutations in patients with known phenotype (HR + HER2- N = 515, HER2 + N = 72, TNBC N = 138). Data are presented as a violin plot with inlayed boxplot, where the middle line is the median, the lower and upper hinges represent the 25th and 75th centiles respectively and the whiskers extend from the hinge to the smallest and largest value, respectively, no further than 1.5 x IQR (interquartile range) from the lower or upper hinge. Data outside of these ranges are plotted individually. Comparison of likely pathogenic mutations with false discovery corrected two-sided pairwise Kruskal-Wallis test, HR + HER2- vs HER2 + q = 0.03, HR + HER2- vs TNBC q < 0.0001. **d** Frequency of copy number increases in ctDNA split by breast cancer subtype in patients with known phenotype (HR + HER2- N = 515, HER2 + N = 72, TNBC N = 138). Comparison with false discovery corrected Chi-squared tests (*MYC* q = 0.01, *PIK3CA* q < 0.0001, *FGFR1* q = 0.01, *EGFR* q = 0.001, *HER2* q < 0.0001, *CCNE1* q < 0.0001, *CCND1* q < 0.001, *CDK6* q = 0.0003, *BRAF* q = 0.0002, *MET* q = 0.01). FS, frameshift; IF, in-frame.

*TP53* in ctDNA (q < 0.0001 and q = 0.0009, respectively, Fig. 2e). Mutations in *BRAF* were identified in ctDNA sequencing in 1.6% (8/515) HR + HER2- cancers, not previously described in tissue sequencing, with activating mutations (G466X, G469X, D594N) and no V600E mutations. Otherwise, the mutational profile identified in ctDNA and tissue across different breast cancer subtypes was broadly similar (Fig. 2e). Microsatellite instability was identified in 1.1% of advanced breast cancers.

Comparing ctDNA-derived mutations to a primary treatment-naive breast cancer tissue sequencing dataset (TCGA)[29], substantial differences were observed (Supplementary Fig. 6)[11,13]. *PIK3CA* mutations were less common in metastatic HR + HER2- disease (q < 0.0001), whilst *ESR1*, *AKT1* and alterations in the epigenetic regulator *ARID1A* were enriched (q < 0.0001, q = 0.006 and q = 0.04, respectively, Supplementary Fig. 6) compared to

primary. The ctDNA genomic profile of advanced TNBC was similar to that of primary TNBC tissue sequencing.

**Clinical and pathological genomic associations and HER2 mutations in HER2 positive breast cancer.** Using the rich clinical trial data available, we explored the clinical and pathological associations of ctDNA mutations, and the maximum variant allele frequency (mVAF) as a proxy of ctDNA purity (Fig. 3a). The number of lines of treatment was associated with increased number of SNVs/indels and mVAF (Fig. 3a), and soft tissue/nodal disease with lower mVAF (13.2 vs 8.0, q = 0.002, Fig. 3a). Patients without a ctDNA alteration were significantly more likely to have had fewer lines of treatment (p = 0.02, Supplementary Tab. 3). *HER2* alterations were more common in

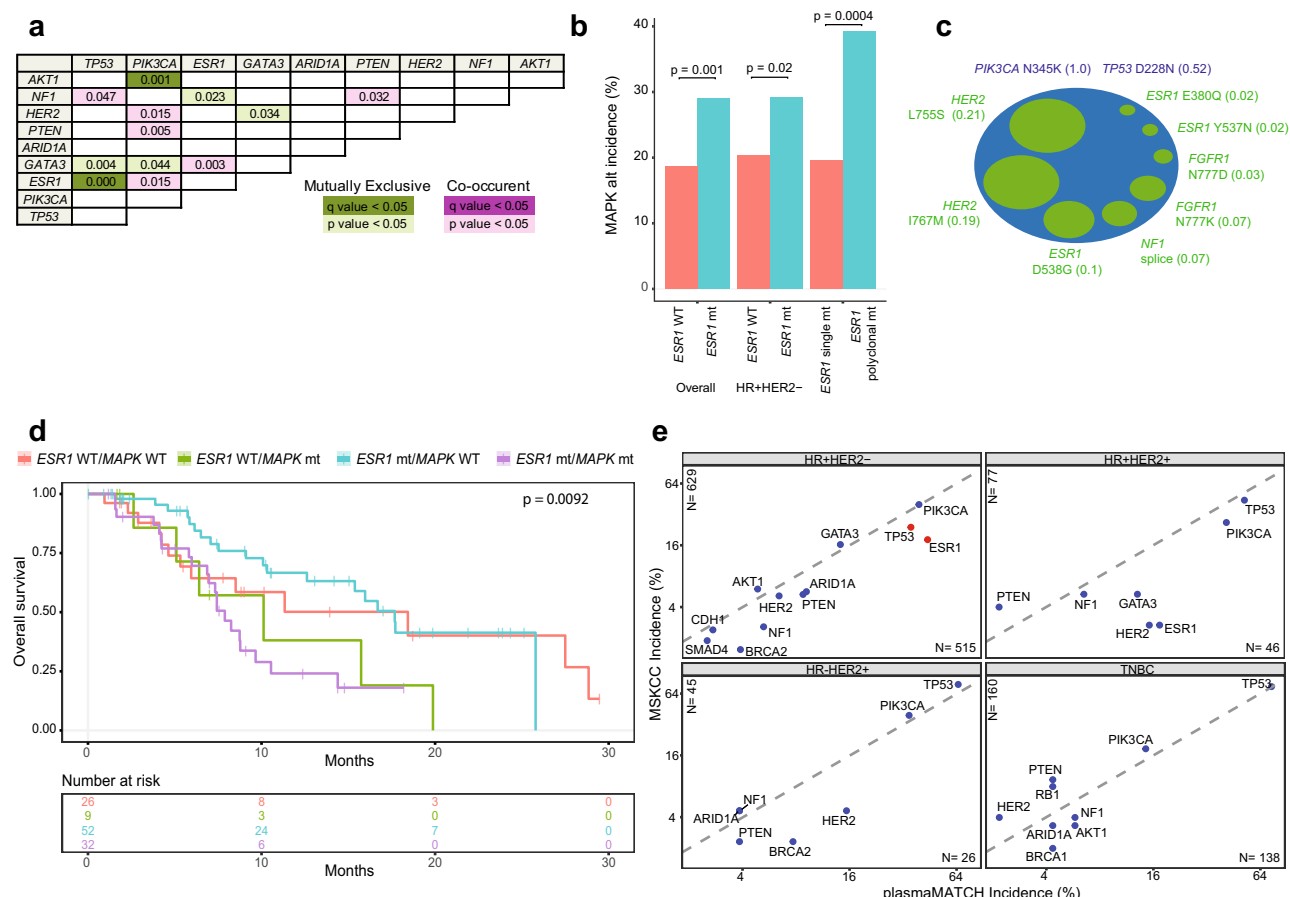

**Fig. 2 Polyclonal resistance with co-enrichment of MAPK pathway and *ESR1* mutations in advanced breast cancer. a** Association analysis for most frequent mutated genes with overall Fisher's exact test two-sided *p*-values. Green genes showing mutual exclusivity, and purple showing co-occurrence with dark colours indicating significance following false discovery correction. **b** Frequency of MAPK pathway alterations comparing *ESR1* mutant (77/265) vs *ESR1* wild-type cancers overall (100/535) (*left*), *ESR1* mutant (66/226) vs wild-type (59/289) in HR + HER2- cancers (*middle*), and within *ESR1* mutant cancers between patients with single (27/138) and polyclonal *ESR1* mutations (50/127) (*right*). *p*-values from two-sided Fisher's exact test. **c** Example of polyclonal genomic resistance in a patient with multiple MAPK pathway and *ESR1* mutations in ctDNA. Blue indicate dominant mutations with cancer fraction, and green subclonal mutations with cancer fraction. **d** Overall survival (OS) in patients with HR + HER2- disease who entered a treatment cohort in plasmaMATCH divided by combined *ESR1* and MAPK pathway mutation status. *ESR1* WT and MAPK WT, median 18.5 months, hazard ratio (HR) -. *ESR1* mt and MAPK WT, median 17.7 months, HR 0.82, 95% confidence interval (CI) 0.40 to 1.69. *ESR1* WT and MAPK mt, median 10.1 months, HR 1.65, 95% CI 0.56 to 4.88. *ESR1* mt and MAPK mt, median 7.9 months, HR 1.65, 95% CI 0.84 to 3.23. *p*-value from log-rank test. HR > 1 indicate worse OS for that group. WT, wild-type; mt, mutant. **e** Mutational profile of ctDNA in plasmaMATCH (*N* = 725 patients with known breast cancer subtype) compared to published large metastatic breast cancer tissue sequencing dataset (MSKCC, *N* = 715 patients with known breast cancer subtype)[11]. Red dots indicate significant change in frequency after false discovery adjusted two-sided Fisher's exact test (HR + HER2-: *ESR1 q* < 0.0001, *TP53 q* = 0.0009). Included are genes with an incidence 1.5% in both data sets.

lobular than in ductal breast cancer (Fig. 3b, *p* < 0.0001)[32,33]. *HER2* mutations were found more commonly in HER2 + cancers with increasing lines of *HER2* directed therapy (*p* = 0.04, Fig. 3c), suggesting acquisition of *HER2* mutations as a mechanism of resistance to prior *HER2* targeted therapies, and identifying a potential novel treatment strategy for HER2 + resistant disease. We validated rare and hotspot *HER2* mutations calls by ddPCR, with 91.7% (22/24) of mutations validating (Supplementary Fig. 7). *RB1* mutations were modestly enriched in patients with HR + HER2- disease and prior CDK4/6 inhibitor exposure, with no evident differences in genomic profile post mTOR inhibition with everolimus (Supplementary Fig. 8)[34]. Patients with tissue HER2 negative disease (*N* = 605) had a significantly lower mean adjusted *ERRB2* copy number in ctDNA compared to those with tissue HER2 positive disease (2.2 vs 9.9, *p* < 0.0001, Fig. 3d). A plasma *HER2* copy number threshold of >2.0 had a sensitivity of 50% (95% CI 37.92–62.08) and specificity of 98% (95% CI 96.77–98.98) in identifying the tissue-based HER2 status defined

on the most recent tissue sample—advanced or archival primary if not available (Fig. 3d).

Alterations within *TP53*, *GATA3*, *ESR1* and *PIK3CA* showed a tendency for organotropism (Fig. 4a). Bone disease was positively associated with *ESR1* and *GATA3* alterations (*q* < 0.0001 and *q* = 0.0009, respectively), and *TP53* negatively associated (*q* = 0.02). Liver disease showed a positive association with *ESR1* alterations (*q* < 0.0001) but a negative association with *TP53* alterations (*q* = 0.002). In HR + HER2- disease *ESR1* mutations were positively associated with liver and bone disease (*q* = 0.004 and *q* = 0.02, respectively, Fig. 4b). After correction for multiple testing, no genes demonstrated a significant pattern of organotropism in TNBC (Fig. 4b).

**APOPEC mutational signature in subclonal mutations of ER positive breast cancer.** To investigate the clonal dominance of individual mutations, we calculated the cancer fraction of each

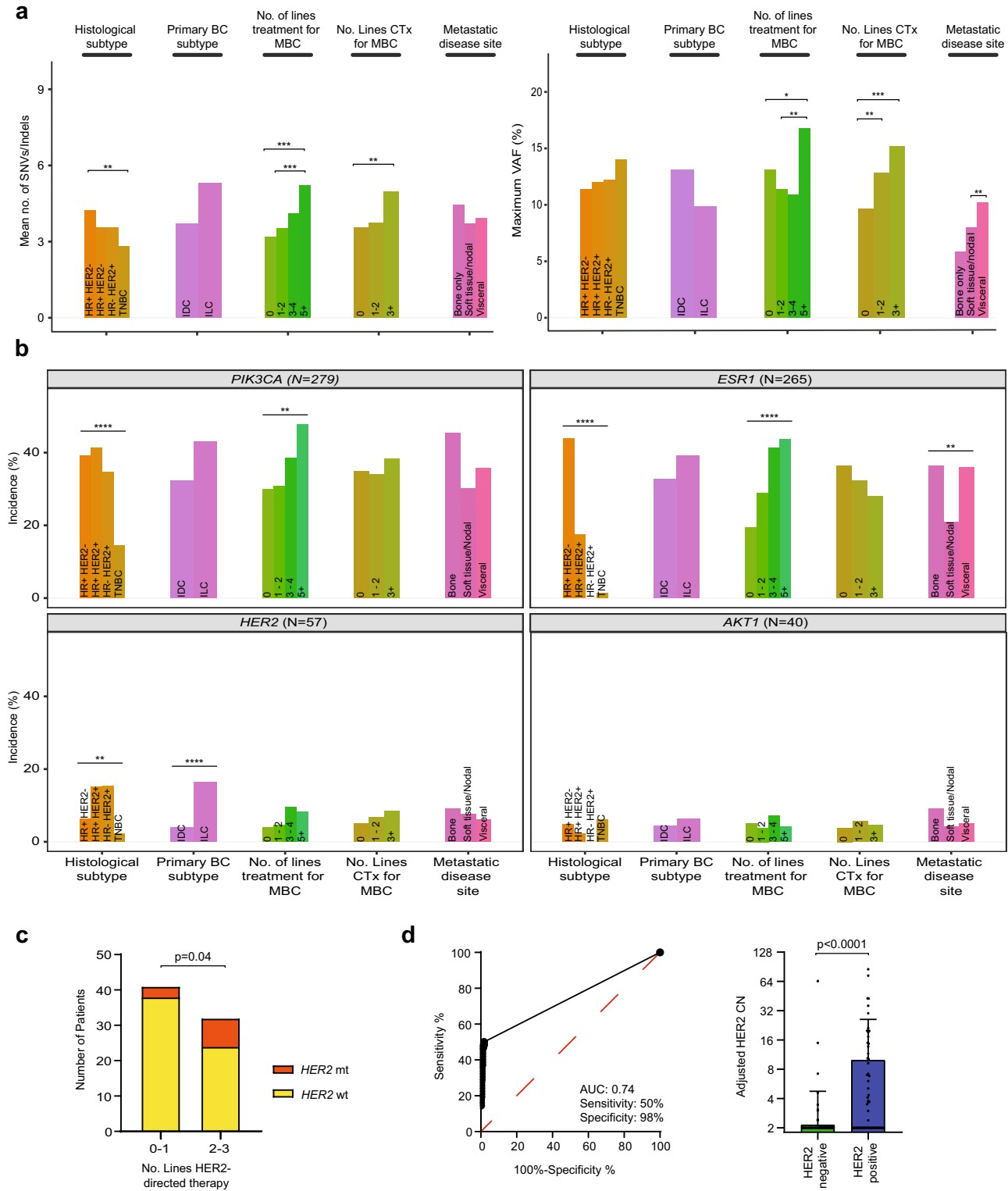

alteration ("Methods", Fig. 5a). Mutations within *AKT1, PIK3CA* and *GATA3* were significantly more likely to be dominant (*q* < 0.0001, *q* < 0.0001, *q* = 0.0003, respectively), while alterations in *ESR1, SMAD4* and *KRAS* were significantly more likely to be subclonal (*q* < 0.0001, *q* = 0.02 and *q* < 0.0001, respectively). *ESR1* mutations were frequently polyclonal (50%, Fig. 5b), occurring almost exclusively in trans (93%, Supplementary Fig. 9). The proportion of dominant to subclonal mutations did not

significantly alter with time from diagnosis of primary breast cancer (Supplementary Fig 10).

To investigate the mutational processes promoting diversity in advanced breast cancer, we aggregated dominant and subclonal alterations by subtype, and performed bootstrap mutational signature analysis ("Methods"). In HR + HER2- disease, subclonal alterations were substantially enriched in signature 13 (APOBEC-related signature). Signature 3 (homologous recombination deficiency) was

**Fig. 3 Clinical and pathological associations of breast cancer mutation profile. a** Association of number of mutations (SNVs/indels, *left*) and the maximum variant allele frequency (mVAF, *right*, as a proxy of ctDNA purity) with indicated clinical and pathological features. p values from pairwise two-sided Kruskal–Wallis test with correction for multiple testing (number of mutations: HR + HER2- vs TNBC $q = 0.008$; 0 vs ≥5 lines of treatment $q = 0.0005$, 1–2 vs ≥5 lines of treatment $q = 0.0005$; 0 vs ≥3 lines of chemotherapy $q = 0.003$. mVAF: 0 vs ≥5 lines of treatment $q = 0.03$, 1–2 vs ≥5 lines of treatment $q = 0.006$; 0 vs 1–2 lines of chemotherapy $q = 0.003$, 0 vs ≥3 lines of chemotherapy $q = 0.0003$; soft tissue/nodal vs visceral disease $q = 0.002$). MBC, metastatic breast cancer; CTx, chemotherapy; IDC, invasive ductal carcinoma; ILC, invasive lobular carcinoma. **b** Association of clinical and pathological features with pathogenic alterations in the four targetable genes in plasmaMATCH: *PIK3CA, ESR1, HER2* and *AKT1*. p-values from Chi-squared test (*PIK3CA*: histological subtype $p < 0.0001$, lines of treatment $p = 0.006$; *ESR1*: histological subtype $p < 0.0001$, lines of treatment $p < 0.0001$, disease site $p = 0.003$; *HER2*: histological subtype $p = 0.004$, primary breast cancer subtype $p < 0.0001$). **c** *HER2* mutation incidence in patients with HER2 + cancer, by line of therapy. 0–1 lines of therapy mutation incidence 7.3% (3/41) and 2–3 lines of therapy mutation incidence 25% (8/32) *HER2* mutations, $p = 0.04$, Chi-squared test. mt, mutant; wt wild-type. **d**) Adjusted *HER2* copy number (CN) in targeted sequencing, in patients with tissue assessed HER2 + (amplified, N = 72) and HER2- (non-amplified, N = 605) cancers. (*left*) receiver operator curve of adjusted *HER2* plasma copy number, (*right*) *HER2* plasma copy number adjusted for purity. Data are presented as mean + SD. The p-value indicated is derived from a two-sided Mann–Whitney U-test.

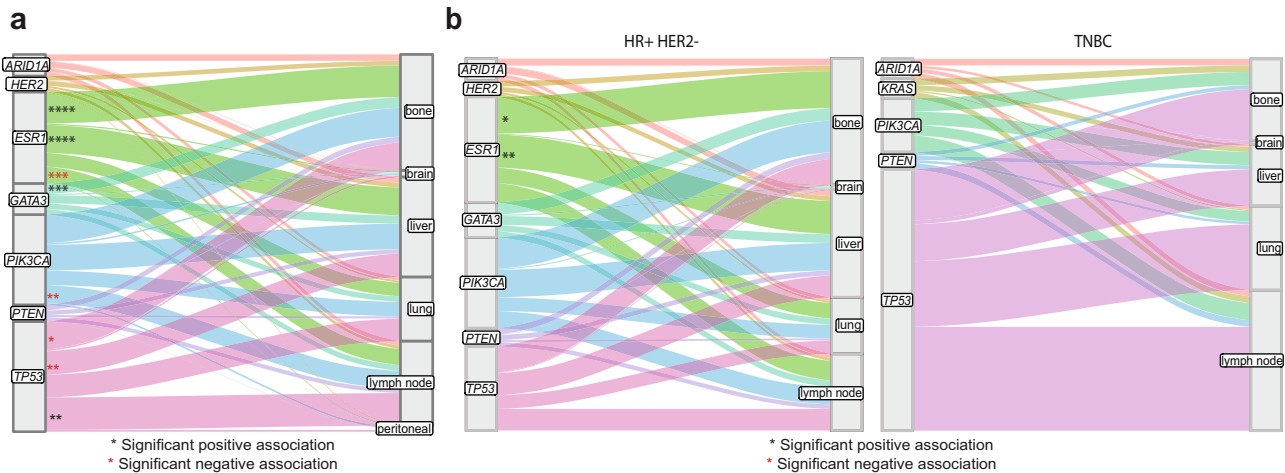

**Fig. 4 Organotropism of mutations in advanced breast cancer. a** Association of mutations in indicated genes with sites of metastasis. p-values from false discovery corrected two-sided Fisher's exact test (*ESR1*: bone $q < 0.0001$, liver $q < 0.0001$, lymph node $q = 0.0001$; *GATA3*: bone $q = 0.0009$; *PIK3CA*: lymph node $q = 0.005$; *TP53*: lymph node $q = 0.002$, liver $q = 0.002$, bone $q = 0.02$). **b** Association of mutations in indicated genes with sites of metastasis in *left* HR + HER2- and *right* TNBC. p-values from false discovery corrected two-sided Fisher's exact test (HR + HER2- *ESR1*: liver $q = 0.004$, bone q = 0.02).

less common in subclonal mutations. In contrast, in TNBC disease subclonal alterations were substantially enriched in age related signature 1 (Fig. 5c). Similarly, subclonal mutations in HR + HER2 + disease were enriched for signatures 2 and 13, whilst subclonal mutations in HR-HER2 + disease were enriched for age related signature 1 (Supplementary Fig. 11). A second mutational signature analysis package designed specifically for targeted sequencing data[35] was utilised to corroborate findings, demonstrating broadly similar results (Supplementary Fig. 12).

**PIK3CA double mutations at APOBEC mutagenesis sites.** Within individual genes, *ESR1* and *PIK3CA* hotspot mutations showed significant variation in clonal dominance (both $p < 0.0001$, Fig. 6a). *ESR1* mutations D538G and Y537S were more dominant than other *ESR1* mutations. *PIK3CA* mutations H1047R/L, N345K and G1049R were dominant, whilst classical hotspot mutations E545K and E542K were of lower clonal dominance, and multiple novel *PIK3CA* mutations were highly subclonal (Fig. 6a and Supplementary Fig. 13a). We validated the novel subclonal *PIK3CA* mutations calls by plasma ddPCR, with 80.0% of mutations revalidating (Supplementary Fig. 13b). Novel *PIK3CA* mutations occurred almost exclusively in the presence of other hotspot mutations, which were frequently present at substantially high clonal fractions (Fig. 6b and Supplementary Fig. 14), suggesting acquisition of second *PIK3CA* mutations.

In HR + HER2- *PIK3CA* mutant disease, 23% (47/202) of patients had multiple *PIK3CA* mutations (Supplementary Fig. 15). The second *PIK3CA* mutation was frequently subclonal, displaying single nucleotide substitutions and mutation contexts consistent with those of APOBEC mutagenesis (Fig. 6c, d). The same predisposition did not occur in *PIK3CA* mutant TNBC disease, where dominant and subclonal alterations were equally likely to occur at APOBEC sites (Fig. 6d). In plasmaMATCH, patients with *ESR1* mutations in ctDNA were enroled in a cohort of extended dose fulvestrant, an oestrogen receptor down-regulator (Supplementary Fig. 1). Patients with multiple *PIK3CA* mutations had worse progression free survival on fulvestrant compared to patients with a single *PIK3CA* mutation or *PIK3CA* wild-type (Fig. 6e).

## Discussion

Utilising the largest prospective ctDNA genomic profiling study of patients progressing with advanced breast cancer, we identify substantial novel features of advanced breast cancer with ctDNA sequencing, demonstrating the ability of ctDNA analysis to dissect spatial heterogeneity and subclonal sampling. In HR + HER-breast cancer ctDNA analysis demonstrates divergent routes to endocrine resistance in individual patients, suggesting that different metastases may develop divergent mechanisms of resistance. Prior advanced tissue sequencing studies demonstrated mutual exclusivity of *ESR1* mutations and MAPK pathway

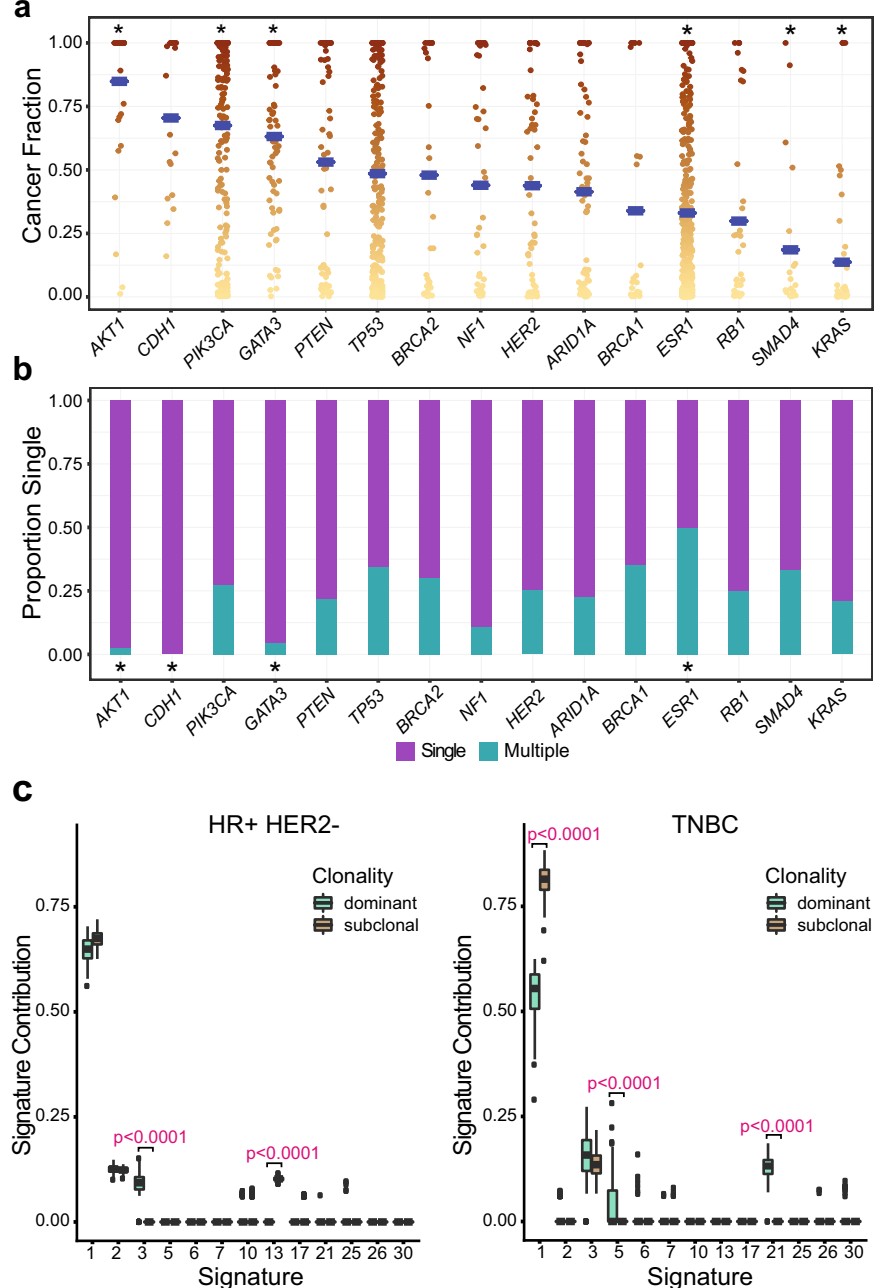

**Fig. 5 Clonal dominance and mutational signatures in dominant and subclonal mutations. a** Cancer fractions of mutations in indicated genes, ordered by mean cancer fraction ($N = 1974$ mutations with assessable cancer fractions). The mean value is indicated with a blue line. *indicates significant difference in cancer fraction compared to remaining cases, false discovery corrected two-sided Wilcoxon signed-rank test (*AKT1* $q < 0.0001$, *PIK3CA* $q < 0.0001$, *GATA3* $q = 0.0003$, *ESR1* $q < 0.0001$, *SMAD4* $q = 0.02$, *KRAS* $q < 0.0001$). Cancer fraction—allele fraction of the mutation relative to the maximum somatic allele fraction in the sample. **b** Proportion of mutations that occur as a single versus multiple mutations per patient in indicated genes. *indicates significant difference in proportion to single to multiple mutations in the gene compared to remaining cases, false discovery corrected two-sided Fisher's exact test (*AKT1* $q = 0.0009$, *CDH1* $q = 0.05$, *GATA3* $q < 0.0001$, *ESR1* $q < 0.0001$). **c** Bootstrap mutational signature analysis on aggregated mutations from all HR + HER2- (*left*, clonally dominant mutations $N = 328$, subclonal mutations $N = 968$) and TNBC (*right*, clonally dominant mutations $N = 121$, subclonal mutations $N = 190$) breast cancers, for dominant and subclonal mutations. Signature contributions for clonal versus subclonal alterations were ascertained using deconstructSigs and compared using a two-sided Mann–Whitney U-test. Signatures with significant difference in signature contribution and no overlap in interquartile range are identified with the p-value (HR + HER2-: signature 3 $p < 0.0001$, signature 13 $p < 0.0001$; TNBC: signature 1 $p < 0.0001$, signature 5 $p < 0.0001$, signature $p < 0.0001$).

alterations in individual metastatic sites[11]. In contrast, we demonstrate *ESR1* mutations co-existing with MAPK pathway alterations (Fig. 2b), in particular in patients with polyclonal *ESR1* mutations (Fig. 2b, c), with polyclonal resistance associated with poor overall survival (Fig. 2d). We report the sub-clonal structure

of advanced breast cancer, and whilst many classic cancer driver genes are shown to be dominant in the cancer (present in all or most tumour cells), multiple other cancer genes are frequently subclonal. We further demonstrate that even within individual genes, such as *ESR1* and *PIK3CA*, different mutations vary in

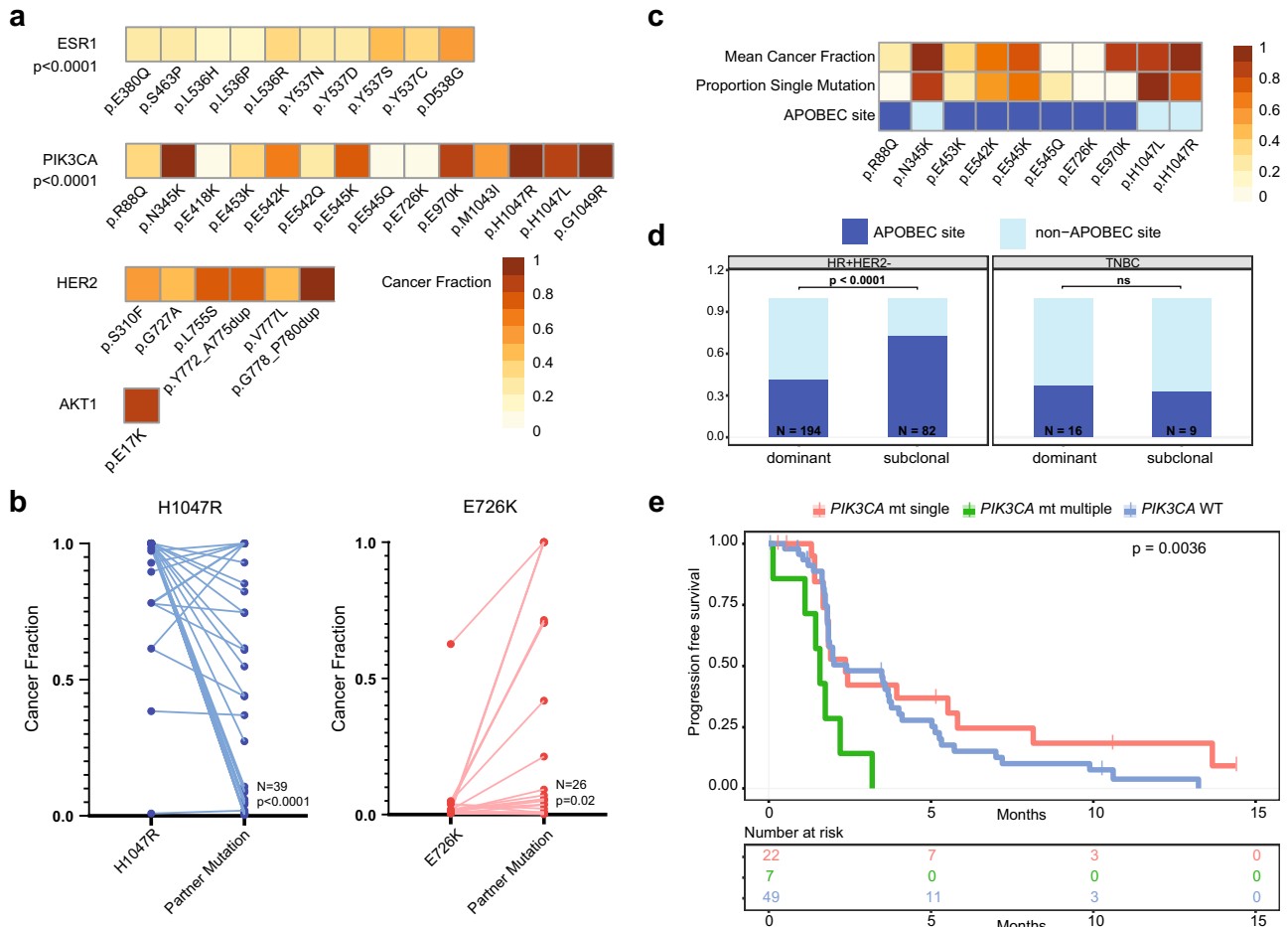

**Fig. 6 Subclonal multiple *PIK3CA* mutations and resistance to fulvestrant. a** Cancer fractions of individual pathogenic hotspot mutations in indicated gene, including hotspots with at least 3 mutations in the overall data set or for any indel. *p*-value from two-sided Kruskal–Wallis test for variation in cancer fraction across mutations in gene (*ESR1* p < 0.0001, *PIK3CA* p < 0.0001). **b** Analysis of patients with H1047R (*left*, N = 39) and E726K (*right*, N = 26) dual pathogenic *PIK3CA* mutations, with linkage of cancer fraction of indicated *PIK3CA* mutation with the other *PIK3CA* mutation present in the same patient. The *p*-values indicated are derived from two-sided Mann–Whitney *U*-tests (H1047R p < 0.0001, E726K p = 0.02). **c** Analysis of individual recurrent hotspot mutations in HR + HER2- *PIK3CA* mutant disease (N = 202) with mean cancer fraction, proportion of mutations detected as a single *PIK3CA* mutation, and indication of whether the mutation occurs at an APOBEC consensus site. Mutations occurring at least 3 times in the HR + HER2- disease dataset included. Cancer fraction and proportion single mutations is lower at APOBEC sites, P < 0.001, two-sided Fisher's exact test both comparisons. **d** Proportion of *PIK3CA* mutations that occur at APOBEC consensus sites, by cancer subtype (HR + HER2- N = 197, TNBC N = 21), and clonally dominant (N = 194 mutations within HR + HER2- breast cancers and N = 16 within TNBC breast cancers) versus subclonal *PIK3CA* mutation (N = 82 mutations within HR + HER2- breast cancers and N = 9 within TNBC breast cancers). *P*-value from two-sided Fisher's exact test. ns, not significant. **e** Progression free survival (PFS) in patients on fulvestrant in treatment cohort A in plasmaMATCH divided by *PIK3CA* mutation status. Cohort A patients with available sequencing data included (78/84). *PIK3CA* WT, median 2.4 months, HR -. *PIK3CA* single mt, median 2.4 months, HR 0.71, 95% CI 0.41 to 1.22. *PIK3CA* multiple mt, median 1.6 months, HR 3.15, 95% CI 0.88 to 11.33. *p*-value from log-rank test. HR > 1 indicates worse PFS for that group. WT, wild-type; mt, mutant.

clonal dominance with some mutations being highly dominant whilst others are frequently subclonal.

We identify a number of new therapeutic approaches for advanced breast cancer. We identify acquired genomic changes in advanced HER2 + breast cancer, with *HER2* mutations selected subclonally though increasing lines of HER2 directed therapies (Fig. 3c). This finding suggests a novel mechanism of resistance to HER2 directed therapy in HER2 positive breast cancer, emphasising the need to investigate whether these cancers would benefit specifically from HER2 tyrosine kinase inhibitors that inhibit mutant *HER2*[36]. In HR + HER2- breast cancer we identify *BRAF* mutations previously not described in advanced breast cancer, with emergence of known activating *BRAF* mutations identified in lung cancer (G466X, G469X, D594N)[37] and no V600E mutations. A small subset (1.1%) of advanced breast cancers have

microsatellite instability in ctDNA sequencing, potentially identifying patients that may benefit from immune checkpoint antibodies[38]. We also show that *HER2* amplification can be identified with high specificity in ctDNA, although sensitivity remains limited favouring recurrent disease biopsy for repeat HER2 testing to identify the small minority of patients who acquire *HER2* amplification at relapse. However, for patients who have disease sites that are not suitable for recurrent disease biopsy, ctDNA testing may present an opportunity to screen for acquisition of *HER2* amplification (Fig. 3d). Detection of *HER2* amplification in plasma in other tumour types has identified responders to HER2 directed therapy[39,40].

Different mutational processes drive diversity in breast cancer subtypes. HR + advanced breast cancer is characterised by subclonal mutations in part generated by APOBEC mutagenesis. We

hypothesise that APOBEC is activated during prior endocrine therapy for advanced cancer, and that this may edit *PIK3CA* to generate frequent second hit novel mutations[12], resulting in hyperactivation of PI3K signalling[41] and resistance to fulvestrant endocrine therapy (Fig. 6e). We performed a mutational signature analysis to differentiate drivers of clonal and subclonal disease. Relative to the broad sequencing approaches of whole-exome and whole-genome sequencing, targeted panels sequence selected areas of interest and as such cover less of the genome, which may limit mutational signature analysis on this data. However, our finding of APOBEC mutations in HR + HER2- subclonal mutations concurs with tissue biopsy sequencing studies[12,15], and confirms that this process contributes to the subclonal diversity of HR + advanced breast cancer. *PIK3CA* mutations vary in clonal dominance (Fig. 6a), and future research will need to investigate whether this variation in subclonality effects response to PI3 kinase inhibitors in the clinic[42]. The poly-clonal nature of endocrine resistance likely substantially challenges attempts to treat endocrine resistant disease. Taken together, our findings emphasise the importance of investigating upfront combination approaches to prevent endocrine resistance. Such approaches could possibly include APOBEC or PI3K pathway inhibitors.

In TNBC, we identified that subclonal diversity is associated with age related mutational signatures, suggesting a lack of specific processes driving subclonality, and potentially suggesting early diversification of metastatic TNBC. We note that patients with TNBC in this series were relatively infrequently treated with targeted therapies or immunotherapy, and it is possible that in the future specific mutational processes selected by targeted therapies will shape subclonality in TNBC. Given the limitations of mutational signature analysis undertaken in targeted sequencing data, these findings require corroboration with whole-exome or whole genome sequencing data.

There are important limitations to our study and considerations to make when understanding the utility of ctDNA in clinical practice. Matched germline blood was not simultaneously sequenced in our study. A small subset of mutations we report in *TP53* and *KRAS* might have originated from clonal hematopoesis as opposed to the cancer[43]. To confidently exclude clonal hematopoesis future research could involve paired germline sequencing, or stringent criteria for variant identification in genes affected by clonal hematopoesis.

Consideration should also be given to the likelihood of false negative results, with some patient groups less likely to shed ctDNA (Fig. 3a), such as those with low burden nodal disease with fewer lines of therapy, where ctDNA may not fully characterise the mutations present. Our study also emphasises that copy number detection is of limited sensitivity in plasma, and for tumour types where copy number events dominate tumour biology alternative approaches of genotyping are required[29]. More comprehensive approaches to genotyping ctDNA such as whole exome sequencing could extend our observations. However, to detect subclonal mutations such approaches will still require error correction, and such approaches are likely to be substantially expensive and likely beyond routine clinical application.

Our findings illustrate the substantial clinical and research potential of ctDNA analysis in defining clonal architecture in cancer, identifying subclonal resistance mutations not appreciable by single site metastatic tumour biopsies, establishing patterns of clonal dominance and characterising the mutational processes that drive diversification of metastatic breast cancer.

## Methods

### Patient consent and blood sampling.
Patients were enrolled prospectively to the plasmaMATCH trial (NCT03182634). plasmaMATCH was co-sponsored by the

Institute of Cancer Research and the Royal Marsden National Health Service (NHS) Foundation Trust, London, UK, and approved by a Research Ethics Committee (16/SC/0271). All participants gave written informed consent before registration for ctDNA testing. ctDNA testing was undertaken using two orthogonal techniques, droplet digital PCR (ddPCR) and error corrected targeted sequencing. Positive mutation status allowed entry into one of five treatment cohorts with a matched targeted therapy (Supplementary Fig. 1a). Patients were eligible to enrol for ctDNA testing within plasmaMATCH if they had advanced breast cancer (ABC) with measurable disease, had progressed on prior therapy for ABC, or relapsed within 12 months of adjuvant chemotherapy, and (following an amendment partway through the trial) had not had more than two lines of chemotherapy for ABC (Supplementary Fig. 1a). Breast cancer subtype was defined using standard clinical ER, PR and HER2 testing on the most recent tissue sample (metastatic, or if not available, the archival primary).

ctDNA testing was undertaken prospectively within the trial by droplet digital PCR (ddPCR) (Supplementary Fig. 1b). Partway through the trial, ctDNA testing was also undertaken prospectively by targeted sequencing, in parallel to ddPCR. For patients enroled prior to prospective targeted sequencing, a banked plasma sample was sent for retrospective testing, where available. This banked plasma sample was either remaining plasma banked after prospective ctDNA screening, or a cycle 1 day 1 (C1D1) pre-treatment plasma sample.

Blood samples for ctDNA testing were collected as described following enrolment into the ctDNA testing component of the plasmaMATCH trial[44]. 30–40 ml of blood was collected in 3–4 10 ml cell-free DNA BCT Streck tubes. 30 ml of blood was shipped at ambient temperature to a central laboratory (Centre for Molecular Pathology, Royal Marsden Hospital) for ddPCR testing. In addition, following protocol amendment, 10 ml were shipped to Guardant Health (Redwood City, California, USA) for targeted sequencing. At the central laboratory, blood samples were centrifuged at $1600\,g$ for 10 min prior to plasma isolation. This was followed by a second centrifuge at $1600\,g$ for 10 min, following which the plasma was aliquoted and stored at $-80\,^{\circ}$C until further analysis. For samples shipped to Guardant Health, samples were centrifuged at $1600\,g$ for 10 min, and the resulting supernatant further centrifuged at $3220\,g$ for 10 min. Isolated plasma was stored at $2\,^{\circ}$C for immediate extraction, or stored at $-80\,^{\circ}$C.

For patients who underwent retrospective targeted sequencing, remaining banked plasma from the ctDNA testing or C1D1 time points was thawed at room temperature. Tubes were inverted 10 times, before a minimum of 2 ml was aliquoted into a separate 4.5 ml cryovial. Aliquoted plasma was stored at $-80\,^{\circ}$C until shipment on dry ice to Guardant Health.

### ctDNA testing by targeted sequencing by Guardant360, Guardant Health.
The Guardant360 targeted sequencing panel identifies single nucleotide variants (SNVs), indels, copy number alterations and fusions within protein-coding regions of 73 (version 2.10) or 74 genes (version 2.11) (Supplementary Fig. 1b). Plasma was isolated from whole blood by double centrifugation followed by cell-free DNA extraction according to the manufacturer's instructions (QIAamp Circulating Nucleic Acid Kit, Qiagen) and quantified prior to library preparation. DNA was labelled with non-random oligonucleotide adaptors (IDT, Inc.), enriched by hybrid capture (Agilent Technologies, Inc.), pooled, and sequenced using paired-end synthesis (NextSeq 500 and/or HiSeq 2500, Illumina, Inc.).

### ctDNA targeted sequencing bioinformatic analysis, variant detection and pathogenicity assessment.
The Guardant Health in-house custom Guardant360 pipeline is described elsewhere[45], but in brief, base calls were demultiplexed using bcl2fastq (v2.19), filtered for base quality and aligned to hg19 using BWA-MEM. SNVs, indels, copy number alterations and fusions were detected by comparing base calls to reference training sets.

The sequencing results generated by Guardant Health underwent further bioinformatics analysis at the Institute of Cancer Research, UK. Data were converted to MAF format using vcf2maf (https://github.com/mskcc/vcf2maf) using MSKCC isoform overrides for annotation with VEP version96[46]. Additional likely germline calls based on a combination of VAF frequency around 50%+− 2% and VAF in general population in the Genome Aggregation Database (https://gnomad.broadinstitute.org/) >0.001% were removed from the dataset. Pathogenic calls were identified by the following process: all mutation calls were further annotated with OncoKB[47] and CancerHotspots[48] and cross referenced against the Cosmic database v90[49] to identify recurrently reported mutations. Mutations were classified as pathogenic based on Cancer Hotspots or OncoKB annotations, recurrent mutations in key breast cancer genes (*ESR1, HER2, PIK3CA, EGFR, RB1* and *FGFR2*) or splicing mutations. MAPK pathway genes included mutations in the following genes (*EGFR, HRAS, KRAS, NRAS, ARAF, BRAF, RAF1, MAP2K1, MAP2K2, MAPK1, MAPK3, FGFR1, FGFR2* and *FGFR3*), fusions of *FGFR2* and *FGFR3*, and copy number changes (CN > 3) in the following genes (*BRAF, EGFR, FGFR1, FGFR2, KRAS*). All analyses were based on mutations screened as likely pathogenic with the exception of mutational signature analysis that included all mutations regardless of their pathogenicity.

### ctDNA testing by droplet digital PCR for plasmaMATCH.
Following enrolment into plasmaMATCH, patients had a blood sample drawn which underwent

**Table 1 *PIK3CA* and *HER2* (*ERBB2*) mutations identified within Guardant Health sequencing validated by droplet digital PCR (ddPCR).**

| Gene | Mutation |
| --- | --- |
| PIK3CA | E545K |
| PIK3CA | E542Q |
| PIK3CA | E726K |
| PIK3CA | E545Q |
| PIK3CA | H1047R |
| HER2 | G727A |
| HER2 | E717D |
| HER2 | Q711H |
| HER2 | L786V |
| HER2 | I628M |
| HER2 | D1105N |
| HER2 | S1002N |
| HER2 | L800R |
| HER2 | L800P |
| HER2 | Q1206K |
| HER2 | R1153Q |
| HER2 | E1079K |
| HER2 | V777M |
| HER2 | D769H |
| HER2 | D769Y |
| HER2 | L755S |
| HER2 | I767M |
| HER2 | S310F |

prospective ctDNA testing for hotspot mutations in *PIK3CA, ESR1, HER2* and *AKT1* (Supplementary Fig. 1b and Supplementary Table 4, and methodology previously published[44]). DNA was extracted from screening plasma using the automated QiaSymphony platform (Qiagen, Hilden). DNA was quantified by Qubit (Thermo Scientific) before being combined with singleplex or multiplex droplet digital PCR (ddPCR) assays targeting hotspot mutations[44] (Supplementary Table 4), supermix and nuclease free water. The samples were partitioned into droplets using an Automated Droplet Generator (Bio-Rad, Pleasanton) before undergoing 40 cycles of PCR on a thermal cycler. Results were analysed on a QX200 Digital Reader (Bio-Rad, Pleasanton). ddPCR assays were considered positive with a minimum of two positive FAM droplets (mutant) per reaction. Poisson probability was used to calculate allele frequency (AF).

**ctDNA validation testing of *PIK3CA* and *HER2* mutations by droplet digital PCR**. DNA was extracted from screening or pre-treatment C1D1 plasma (to match the timepoint sequenced by targeted sequencing with Guardant360) using the automated QiaSymphony platform (Qiagen, Hilden) or manually using the QIAamp Circulating Nucleic Acid Kit (Qiagen, Hilden) following the manufacturer's instructions. Extracted DNA was quantified by Qubit (Thermo Scientific) or by Taqman assay droplet digital PCR (ddPCR). Following quantification, DNA was combined with singleplex ddPCR assays targeting specific mutations (Table 1 and Supplementary Table 4), supermix and nuclease free water before partition into droplets using an Automated Droplet Generator (Bio-Rad, Pleasanton). Samples underwent 40 cycles of PCR on a thermal cycler before analysis on a QX200 Digital Reader (Bio-Rad, Pleasanton). ddPCR assays were considered positive with one or more positive FAM droplets (mutant) per reaction. Poisson probability was used to calculate allele frequency (AF).

**Cancer fraction**. Variant allele frequency (VAF) is dependent on plasma ctDNA purity, with the number of cancer cells harbouring the mutation relative to all the cancer cells present, and any copy number alteration affecting the gene[50]. With the assumption that the mutation with the highest allele frequency in a sample represents a truncal mutation present in every cancer cell, the clonality of any other mutations present can be calculated relative to the mutation with greatest allele frequency, termed cancer fraction. The cancer fraction was calculated as the allele frequency of the mutation relative to the maximum somatic VAF (mVAF) present in the sample. Samples with a single alteration were not assessable for cancer fraction. Analysis of copy number versus allele frequency revealed that, in this dataset of patients with metastatic breast cancer, copy number changes had negligible influence on allele frequency (Supplementary Fig. 16). This allowed for global comparison across the dataset, nevertheless recognising that for individual

mutations the local copy number status may affect classification. Analyses based on categorical assessment of clonality, mutations with cancer fraction ≥0.5 were considered dominant, whilst mutations with cancer fractions <0.5 were considered subclonal.

**HER2 copy number adjustment**. The plasma *HER2* copy number identified is influenced by the purity, or amount of tumour DNA relative to normal DNA, of the sample. To account for this, we adjusted the copy number relative to the mVAF in the sample using the below formula, as described elsewhere[39]:

$$\text{Adjusted } pCN = [\text{Observed } pCN\text{-}2^*(1 - T\%)]/T\%$$

$$\text{where } T\% = 2X \, mVAFmax/100$$

$$pCN = \text{patient Copy Number}$$

**Comparison of plasmaMATCH dataset with MSKCC and TCGA datasets**. Data for MSKCC[11] dataset was downloaded from cbioportal (https://www.cbioportal.org/). Metastatic breast cancer samples where the patient phenotype was either HR + HER2-, HR + HER2 + , HR-HER2 + , or TNBC were included. If a patient had two samples, the most recent biopsy result was analysed. The TCGA dataset was downloaded from the Genomic Data Commons Data Portal (https://portal.gdc.cancer.gov/). Only primary breast cancer samples where the patient phenotype was either HR + HER2-, HR + HER2 + , HR-HER2 + , or TNBC were included.

Mutation calls from both datasets were annotated using the same ICR bioinformatics pipeline as the plasmaMATCH dataset (described earlier). Mutation calls were filtered to only include loci present in both sequencing panels for comparison and mutations judged to be pathogenic (described earlier). Fisher's exact test was used to compare the proportions of gene alterations within each dataset for each phenotype, with adjustment for multiple testing using the Benjamini-Hochberg procedure (false discovery rate, FDR).

**Mutational signatures**. For HR + HER2- and TNBC respectively, somatic alterations were aggregated into clonal (cancer fraction ≥ 0.5) versus subclonal alterations. Each mutation site was counted once, however, if a mutation was differentially classified as clonal and subclonal in different patients, the mutation was counted in both sets. Signature contributions were estimated using the deconstructSigs R package[51] using a set of 30 mutational signatures from Cosmic[52]. Bootstrap sampling to define confidence intervals of assignments was applied sampling 90% of the data in 200 iterations for each subtype. Signature contributions for clonal versus subclonal alterations were compared using the Mann-Whitney U test. Orthogonal analysis was also carried out using SigMA[35] with tumour type breast cancer using exome trinucleotide counts (Supplementary Fig. 12). For categorical analysis of APOBEC consensus sites the following trinucleotides were included: T(C > G)T, T(C > G)A, T(C > A)[N] on both DNA strands.

**Statistical analyses**. Data was collated in Microsoft Excel (2016). Statistical analysis was carried out using R version 3.5.2 and Graphpad Prism version 8.0.1. Categorical data was analysed using a Chi-squared test for group-wise analysis, and pairwise Fisher's exact test (R package RVAideMemoire) for intergroup comparisons. Where indicated with a q value rather than a p value, a correction has been made for multiple testing using the Benjamini–Hochberg procedure. Continuous data was compared using the Kruskal–Wallis test, and where a significant finding was found a pairwise Kruskal–Wallis test (package pairw.kw() function from R package asbio) was used to identify individual significant differences following correction for multiple testing based on the Bonferroni procedure. Where data was paired the Wilcoxon signed-rank test was utilised. Co-occurrence and mutual exclusivity of gene alterations was assessed using the Fisher's exact test, with correction for multiple testing using the Benjamini–Hochberg procedure. Genes with alterations to an incidence of 5% in the cohort were included. Two-sided p values have been used throughout. Time to event survival data were analysed with log-rank test and hazard ratios were calculated with Cox regression. Plots were created using Microsoft Excel, Graphpad Prism and the R software packages ggplot2, pheatmap, ggalluvial and survminer.

**Reporting summary**. Further information on research design is available in the Nature Research Reporting Summary linked to this article.

## Data availability

The processed plasmaMATCH Guardant360 sequencing data generated and analysed during the current study are available within Supplementary Data 1. We do not have permission from the patients to publicly deposit the raw sequencing data. To protect the privacy and confidentiality of patients in this study, clinical data are also not made publicly available. The data can be obtained by submitting a formal data access request in accordance with the Institute of Cancer Research Clinical Trials and Statistics Unit (ICR-CTSU) data and sample access policy. Requests are to be made via a standard proforma

describing the nature of the proposed research and extent of data requirements which is reviewed by the trial management group. Data recipients are required to enter a formal data sharing agreement, which describes the conditions for data release and requirements for data transfer, storage, archiving, publication, and intellectual property. Trial documentation including the protocol are available on request by contacting plasmamatch-icrctsu@icr.ac.uk.

The MSKCC data are available in the cBioPortal for Cancer Genomics database (https://www.cbioportal.org/study/summary?id=breast_msk_2018).

The TCGA data are available in the cBioPortal for Cancer Genomics database (https://www.cbioportal.org/study/summary?id=brca_tcga_pan_can_atlas_2018). TGCA data analysed for this manuscript were released 28th January 2016.

All other data are available within the Article, Supplementary Information or available from the authors upon request.

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

## Acknowledgements

This research was funded by Cancer Research UK and Breast Cancer Now, and sequencing of ctDNA was conducted by Guardant Health. The plasmaMATCH trial is funded by Cancer Research UK (CRUK/15/010, C30746/A19505), with additional support from AstraZeneca, Puma Biotechnology, Guardant Health and BioRad. Grateful thanks to all trial participants and their families. We thank Breast Cancer Now for

funding this work as part of Programme Funding to the Breast Cancer Now Toby Robins Research Centre. Thanks also to staff at participating centres, the ICR-CTSU trial team, the staff at central laboratories, and Dr Syed Haider and his team in the Breast Cancer Now Toby Robins Research Centre Bioinformatics Core Facility for bioinformatics support. plasmaMATCH is co-sponsored by the Institute of Cancer Research and the Royal Marsden National Health Service Foundation Trust. ICR-CTSU is supported by the Cancer Research UK core grant (C1491/A25351). The trial is registered NCT03182634 and ISRCTN:16945804. plasmaMATCH is supported by the National Institute for Health research (NIHR) Manchester Clinical Research Facility at The Christie Hospital, Manchester UK, and by the Cancer Research UK Cambridge Centre, the Cambridge NIHR Biomedical Research Centre and the Cambridge Experimental Cancer Medicine Centre, Cambridge UK, and the NIHR Biomedical Research Centre at University College London Hospital (UCLH), London UK. plasmaMATCH is supported at participating sites in England by the NIHR Clinical Research Network, in Scotland by the Chief Scientist Office, and in Wales by Health and Care Research Wales. JSR-F is funded in part by the Breast Cancer Research Foundation, by the National Institutes of Health (1 R01 CA244812-01) and by the National Cancer Institute Cancer Center Core Grant (P30-CA008748). This study represents independent research supported by the NIHR Biomedical Research Center at The Royal Marsden NHS Foundation Trust and the Institute of Cancer Research, London. The views expressed are those of the author(s) and not necessarily those of the NIHR or the Department of Health and Social Care. The authors also acknowledge past and present colleagues on the plasmaMATCH trial management group, the independent data monitoring committee, and the trial steering committee, who provided oversight of the trial. This study was presented in part at the 2019 San Antonio Breast Cancer Symposium, December 10-December 14, 2019, San Antonio, Texas (abstract number GS3-07).

## Author contributions

N.C.T., I.G.M., J.M.B., A.M.W., I.R.M., R.D.B., R.R. and A.R. conceived and designed the plasmaMATCH trial. B.K., R.J.C., I.G.M., J.M.B., A.R. and N.C.T. conceived and designed the project. B.K., H.B., M.B., G.W.-C., S.H., C.S., S.K., L.M., K.W., M.H., I.F., K.C.B. and R.B.L. collected and assembled the data. B.K., R.J.C., L.S.K., J.S.R.-F., I.G.M., J.M.B., A.R. and N.C.T. undertook data analysis and interpretation. B.K., R.J.C., I.G.M. and N.C.T. wrote the manuscript.

## Competing interests

N.C.T., AR, JMB, L.S.K., C.S., L.M., S.K., K.W., S.M., H.B., M.H., B.K, I.G.M., M.B., G.W.-C., S.H. and R.C. report grants from Cancer Research UK, grants and non-financial support in the form of study drug provision from AstraZeneca and Puma Biotechnology and non-financial support in the form of ctDNA sequencing from Guardant Health and provision of reagents from BioRad during the conduct of the study. N.C.T. also reports grants and personal fees from AstraZeneca, Pfizer, and Roche/Genentech, personal fees from Bristol-Myers Squibb, Lilly, MSD, Novartis, Bicycle Theraputics, Taiiho, Zeno Pharmaceuticals and Repare Therapeutics and grants from BioRad, Clovis, Merck Sharpe and Dohme, and Guardant Health outside the submitted work. B.K. also reports personal fees from Guardant Health outside the submitted work. A.M.W. reports personal fees from Roche, personal fees and other support from Novartis, Pfizer, Lilly, Daiichi-Sankyo, MSD, AstraZeneca, Athenex and other support from Seattle Genetics, Andrew Wardley Ltd, Manchester Cancer Academy and Outreach Research and Innovation Group Limited outside the submitted work. I.R.M. reports personal fees and non-financial support from Roche Products UK Ltd, Eli Lilly and Eisai and personal fees from Novartis, Pfizer, Daichi Sankyo, Genomic Health, Pierre Fabre and MSD outside the submitted work. R.D.B. reports grants from AstraZeneca and Roche/Genentech outside the submitted work. R.R. reports personal fees from Novartis, Eli-Lilly and Pfizer, personal fees and non-financial support from Daiichi Sankyo and G1Therapeutics and non-financial support from Roche and AstraZeneca outside the submitted work. H.B. also reports personal fees from AstraZeneca outside of the submitted work. M.H. also reports also reports personal fees from Bristol Myers Squibb, Boehringer Ingelheim, Roche Diagnostics and Eli Lilly during the conduct of the study. J.M.B. also reports grants and non-financial support from AstraZeneca, Novartis, Janssen-Cilag, Merck Sharpe & Dohme, Pfizer, Roche, and Clovis Oncology and grants from Medivation outside the submitted work. A.R. also reports personal fees from Roche Products Limited, Pfizer, Novartis, Lilly and M.S.D. outside the submitted work. J.S.R.-F. is a consultant of Goldman Sachs and REPARE Therapeutics, a member of the scientific advisory board of Volition Rx and Paige.AI, and an ad hoc member of the scientific advisory board of Ventana Medical Systems, Roche Tissue Diagnostics, Genentech, Novartis and InVicro. I.F., K.S.B. and R.B.L. are employees with stock ownership in Guardant Health, Inc.
