## [Peer Review File · Nature Communications]

REVIEWER COMMENTS

Reviewer #1 (Remarks to the Author):

The authors report a panel of 74 genes sequenced in plasma cfDNA from ~800 patients in an umbrella trial of matching therapies using cfDNA in advanced breast cancer. The main finding is less than half OS in patients with double MAPK/ESR1 mutations. The report adds to the literature as a resource. There are some superficial/concerning aspects of the analysis/presentation which should be addressed.

Concerns:

The authors are advertising this as a landscape - it certainly is not a landscape, 74 genes with a tiny fraction of the genome covered does not meet the general understanding. They should temper their claims.

Poor resolution graphics make some figures frankly un-reviewable: Fig 2, Fig 4, Fig 6b, Ex fig 9a, are problematic, extended fig 10 – absolutely uninterpretable. Many figure legends are poorly labeled/described. wastes reviewer time trying to decipher them.

Methodological/analytical limitations

(1) A sensitivity analysis of the main conclusions to false positives towards the lower limit of practical detection ~0.1% with this assay is not conducted. Although the assay is claimed in earlier publications to detect mutations down to 0.02-0.04%, recent work (eg Landau lab, Nature Medicine 2020) has suggested that small scale targeted assays struggle below 0.1% where the signal/noise becomes adverse due to the physical limitations. It is notable that in the only ddPCR validation I could see in the manuscript, of PIK3CA non-canonical variants, the accuracy was much lower (83%) compared with previously published figures for this assay at hotspots. The authors should establish more clearly the sensitivity/PPV for the most important gene comparisons with independent validation.

(2) Position/gene specific error model could account for differences between genes towards the lower limits of detection and this could affect many of the analyses of putative subclonal mutations as these would be polluted with false positives. Its notable in extended figure 4 that there are many variants below 0.1 for MAPK but only two for ESR – differences in sensitivity/specificity that is locus specific could account for this. The mutations around the 0.1 detection range should be validated by ddPCR in this case.

(3) The analysis of copy number and allele fraction is in some sense circular since it depends on the same read data. The strange ROC curve for ERBB2 shows only 50% sensitivity to amplifications, which suggests the ability to discriminate important copy number effects will be limited. The consequences of clone specific copy number effects are probably undetectable in this approach.

Assumptions about copy number changes would strongly influence subclonality calculations in the manner presented. The supplemental analysis is not convincing in this regard.

(4) Signature analysis – is very difficult to be certain of the mutational processes involved with limited scope of sequencing, which is also necessarily biased in regions. Other processes such as transcription coupled repair, differences between chemotherapy treatments between patients could contribute. The bootstrap analysis will not magically produce more signal, it can only help with the false discovery rate and the number of positions/mutations is very limited as a function of the assay. The authors don't present a robust quantitative estimation of the nearest neighbor signatures. Figure 5c implies a comparison – unclear what the vertical scale % denominator is, and whether the unequal group sizes of clonal/subclonal and the effect of very different absolute number of mutations were properly accounted for. Possible influences of clonal hematopoiesis resulting in bystander non-tumour cfDNA are not commented on.

Reviewer #2 (Remarks to the Author):

As the utility of circulating tumor DNA (ctDNA) continues to unfold in cancer, it is of certain value to reanalyze the genomic landscape of tumor types where the so called liquid biopsy approach might be adopted clinically. In this study, the authors take advantage of the samples and associated data collected (at baseline, pre-treatment) in the context of the plasmaMATCH trial, with metastatic breast cancer patients undergoing serial ctDNA testing in order to more precisely guide their treatment.

This is an elegant work that “checks all the boxes” in this type of cancer genomics studies (description of the landscape, comparison to previously assembled datasets, study of clinical and pathological associations, etc), with the main novelties being the type of samples and the technology being used. The compendium of resulting observations, not only makes a nice companion paper to the trial main publication, but also offers a few potential new therapeutic approaches for advanced breast cancer, including some that might prevent acquired resistance to certain treatments.

I don't have major issues with the manuscript as it is. However, as the authors acknowledge, there are several important limitations to the study. It would be ideal that they discuss potential avenues to overcome these, like ways to deal with potential clonal hematopoiesis or qualifying samples for their approach by setting a threshold for ctDNA shedding in tumor samples. This could help with better assay development for future prospective studies that will continue to assess the accuracy of ctDNA analysis for routine practice and its potential to guide targeted therapy without requiring solid tissue testing (limited to dissect spatial heterogeneity and subclonal sampling).

Speaking of tissue based sequencing, it would be appreciated if the authors provide some information about how the MSK-IMPACT cohort compares to the one in this trial regarding patient characteristics (for instance, lines of previous treatment).

One comment about the authors observations in TNBC. Given that the ctDNA genomics landscape in this subtype was similar to that of primary tissue sequencing, is it fair to conclude that ctDNA analysis is less informative in this subset of breast cancers? Is it because of the higher number of copy

number alterations in TNBC? Would it be hard to replace tissue biopsy sequencing when it comes to copy number-based biomarkers assessment (as stated in one of the manuscripts), especially in this subtype?

Lastly, could the authors comment on the potential of more comprehensive sequencing approaches (WES) of ctDNA (as an alternative to approaches like the Guardant360 sequencing-based assay) to more accurately profile the mutation landscape of breast cancer patients in the future, including the identification of low frequency mutations, despite the lower read depth and the current lack of standardization of such approaches?

Reviewer #3 (Remarks to the Author):

Authors should be commended on this excellent manuscript describing the plasmaMATCH cohort. Findings are interesting and highly relevant. Comment should be made on patients that had no findings in their plasma. Who are these patients and what should be done with them?

I would also consider bringing Extended Table 1 into the main manuscript as details about the metastatic patient population specifically lines of prior therapy as well as time since Dx to plasma sampling are important. To this end, can the authors comment on the Time categories and if more subclonal/clonal mutations are found with duration?

REVIEWER #1 (REVIEWER COMMENTS TO THE AUTHOR):

Overview:

The authors report a panel of 74 genes sequenced in plasma cfDNA from ~800 patients in an umbrella trial of matching therapies using cfDNA in advanced breast cancer. The main finding is less than half OS in patients with double MAPK/ESR1 mutations. The report adds to the literature as a resource. There are some superficial/concerning aspects of the analysis/presentation which should be addressed.

Concerns:

1) The authors are advertising this as a landscape - it certainly is not a landscape, 74 genes with a tiny fraction of the genome covered does not meet the general understanding. They should temper their claims.

Response: We have altered the word 'landscape' to 'profile' in the manuscript title and throughout the text.

2) Poor resolution graphics make some figures frankly un-reviewable: Fig 2, Fig 4, Fig 6b, Ex fig 9a, are problematic, extended fig 10 – absolutely uninterpretable. Many figure legends are poorly labeled/described. wastes reviewer time trying to decipher them.

Response: We have revised the figures to ensure that all font is Arial, black and at the minimum pre-specified 5pt size. Figure 6b has been revised. High resolution images were supplied, but during PDF generation may have become low resolution. We provide high resolution images at resubmission.

Extended Figure 10: We have revised this image to increase the font size. Gene annotation on this figure is <5pt, but we do not feel this detracts from the message of the figure which is to demonstrate the clonal dominance relationships of dual *PIK3CA* mutations.

We have revised the figure legends throughout the manuscript to ensure comprehensibility.

Methodological/analytical limitations

3) A sensitivity analysis of the main conclusions to false positives towards the lower limit of practical detection ~0.1% with this assay is not conducted. Although the assay is claimed in earlier publications to detect mutations down to 0.02-0.04%, recent work (eg Landau lab, Nature Medicine 2020) has suggested that small scale targeted assays struggle below 0.1% where the signal/noise becomes adverse due to the physical limitations. It is notable that in the only ddPCR validation I could see in the manuscript, of PIK3CA non-canonical variants, the accuracy was much lower (83%) compared with previously published figures for this assay at hotspots. The authors should establish more clearly the sensitivity/PPV for the most important gene comparisons with independent validation.

Response: We understand the concern to be regarding the specificity of variant detection of low allelic fraction variants and the concern that some of these may represent false positives. Our confidence in the validity of the mutation calls and analyses arises from several data sources, as listed below. We also highlight that in the literature the veracity of the sequencing assay use is widely accepted. For example, in the current issue of Nature Medicine (<https://www.nature.com/articles/s41591-020-1063-5>). Extensive validation has been published on the sequencing assay (<https://clincancerres.aacrjournals.org/content/24/15/3539.long>), with the error rates of the assay (for example in Figure 2), making it implausible that the results we present reflect sequencing error¹. In clinical validation experiments, 222 cancer samples with variants detected by Guardant360 underwent ddPCR validation, demonstrating a PPV of 99.6% (VAF 0.1 – 94%) and NPV of 97.8%¹. The assay has in addition presented very substantial validation in the FDA approval of the assay². The FDA approval of the assay as a companion diagnostic to predict response to osimertinib placed no lower limit on the allele fraction of targetable *EGFR* L858R, exon 19 deletion, or T790M mutations, with the latter reported as low as 0.03% with response in the AURA study². Independence of targeted therapy response from low vs. high levels of targetable alterations with the Guardant360 assay already been shown across different classes of targetable genomic alterations with AF as low as 0.06% in multiple earlier outcomes studies^{3,4}.

We provide extensive orthogonal validation in plasmaMATCH. Firstly, the plasmaMATCH trial tested the hotspot mutation status of four genes, *PIK3CA*, *ERBB2*, *ESR1* and *AKT1*, with both digitalPCR and Guardant360 in 784 patients. Data recently published in the companion manuscript⁵ demonstrated a high level of agreement

between the two techniques, with Kappas ranging from 0.89–0.93 for gene-level agreement⁵. For all positive calls for exact mutation status, between the two techniques the agreement level was 77.6%. Therefore, we demonstrate a very high level of agreement with an orthogonal assay, measured in a very high number of patients.

In response to the reviewer’s comment, we have analysed the agreement between mutation calls of plasmaMATCH targetable hotspots within *PIK3CA*, *HER2*, *AKT1* and *ESR1* overall, and for mutations with allele frequency by ddPCR of <1%. At low allele frequencies, the sensitivity of a mutation call by Guardant360 is 80.9%, compared to 90.9% for all mutations (below). At lower allele frequencies, stochastic sampling will reduce sensitivity. We highlight the very high degree of specificity seen at low allele frequencies.

All mutation calls		ddPCR	
		Positive	Negative
Guardant360	Positive	492	77
	Negative	49	9841

	%
Sensitivity	90.9
Specificity	99.2
ppv	86.5
npv	99.5

ddPCR AF<1%		ddPCR	
		Positive	Negative
Guardant360	Positive	161	77
	Negative	38	9841

	%
Sensitivity	80.9
Specificity	99.2
ppv	67.6
npv	99.6

Furthermore, we have validated 20 subclonal *PIK3CA* mutations with digital PCR, with 16/20 (80.0%) mutations validating (Extended data Figure 11b). Given that the majority of *PIK3CA* mutations that underwent ddPCR validation had an allele frequency <1% (mean allele frequency 0.56%), and after considering stochastic effects, this level of validation is remarkably high. Please see the additional validation of *ERBB2* mutations below.

4) Position/gene specific error model could account for differences between genes towards the lower limits of detection and this could affect many of the analyses of putative subclonal mutations as these would be polluted with false positives. Its notable in extended figure 4 that there are many variants below 0.1 for MAPK but only two for ESR – differences in

sensitivity/specificity that is locus specific could account for this. The mutations around the 0.1 detection range should be validated by ddPCR in this case.

Response: As requested, we have re-analysed the data for Figure 5a with a variant cutoff limit of ≥ 0.1 , to address the reviewers concern that allele fractions detected below 0.1 could affect the results. There overall is no change to the results (shown below) demonstrating that our conclusions are not affected by the position/gene specific error model.

Figure 5a. All allele fractions

Figure 5a. Repeated only including allele fractions ≥ 0.1 . There are essentially no meaningful difference to the Figure 5a in the manuscript.

In reviewing Extended data Fig. 4 in response to the reviewers comment, we realized that there was an error in the original figure that raised the reviewers concern. In the originally submitted figure, summed allele fractions for polyclonal *ESR1* mutations

were shown that explained the lack of low allele fraction *ESR1* mutation. We have now revised as below, to correct this error.

Extended data Fig. 4: Correlation of *ESR1* and *MAPK* pathway mutation allele fractions from the same patient

Correlation of allele fraction of *ESR1* and *MAPK* pathway mutations from the same patient, in patients with single *ESR1* and *MAPK* mutations. Spearman correlation coefficient -0.264 , $p=0.017$. Includes all patients with mutations in both with HR+HER2-disease.

As an additional action undertaken in response to the reviewer’s comment, to validate *MAPK* pathway activating mutations, we have validated hotspot and rare *HER2* alterations, with 91.7% (22/24) validating. We have included this data as Extended data Figure 6, and have stated on page 7:

“We validated rare and hotspot *HER2* mutations calls by ddPCR, with 91.7% (22/24) of mutations revalidating (Extended data Fig.6).”

Extended data Fig. 6: Validation of rare and hotspot mutations in *HER2* by digital PCR

Association between allele frequency in ctDNA sequencing and validation analysis with plasma DNA digital PCR (ddPCR), n=24. 22/24 (91.7%) mutations were validated by ddPCR. Spearman correlation coefficient 0.76, $P<0.0001$. ND, not detected.

5) The analysis of copy number and allele fraction is in some sense circular since it depends on the same read data. The strange ROC curve for ERBB2 shows only 50% sensitivity to amplifications, which suggests the ability to discriminate important copy number effects will be limited. The consequences of clone specific copy number effects are probably undetectable in this approach. Assumptions about copy number changes would strongly influence subclonality calculations in the manner presented. The supplemental analysis is not convincing in this regard.

Response: We agree with the reviewer that copy number analysis in ctDNA is limited by the inability to differentiate clone-specific differences in copy number. The ROC curve simply demonstrates that detection of any level of *HER2* amplification above background in ctDNA sequencing signifies the presence of *HER2* amplification in the tumor. This is highly valuable confirmation, although we entirely agree with the reviewer's comment that sensitivity is limited at 50%.

We highlight the poor sensitivity of ctDNA copy number analysis to identify *HER2* amplification in the discussion, on page 10, as follows:

“We also show that HER2 amplification can be identified with high specificity in ctDNA, although sensitivity remains limited favouring recurrent disease biopsy for repeat HER2 testing to identify the small minority of patients who acquire HER2 amplification at relapse. However, for patients who have disease sites that are not suitable for recurrent disease biopsy, ctDNA testing may present an opportunity to screen for acquisition of HER2 amplification (Fig. 3d).”

We do show that copy number changes do not broadly affect the allele fraction of mutations detected in plasma (Extended data Fig. 14). The reviewer states that this data is ‘not convincing’, but we respectfully disagree. There is clearly no association between copy number and allele fraction, strongly suggesting that copy number does not broadly impact allele fraction. We of course recognize that for some individual events this may not hold true, and this is discussed appropriately in the methods section on page 16, as follows:

“Analysis of copy number versus allele frequency revealed that, in this dataset of patients with metastatic breast cancer, copy number changes had negligible influence on allele frequency (Extended data Fig. 14). This allowed for global comparison across the dataset, nevertheless recognising that for individual mutations the local copy number status may affect classification.”

6) Signature analysis – is very difficult to be certain of the mutational processes involved with limited scope of sequencing, which is also necessarily biased in regions. Other processes such as transcription coupled repair, differences between chemotherapy treatments between patients could contribute. The bootstrap analysis will not magically produce more signal, it can only help with the false discovery rate and the number of positions/mutations is very limited as a function of the assay. The authors don’t present a robust quantitative estimation of the nearest neighbor signatures. Figure 5c implies a comparison – unclear what the vertical scale % denominator is, and whether the unequal group sizes of clonal/subclonal and the effect of very different absolute number of mutations were properly accounted for. Possible influences of clonal hematopoiesis resulting in bystander non-tumour cfDNA are not commented on.

Response: We apologise that Figure 5c was unclear and have altered the labelling on the plots to reflect the vertical scale and added number of mutations for each subtype. The legend has been altered to explain the tests applied:

“c) Bootstrap mutation signature analysis on aggregated mutations from all HR+HER2- (left, clonally dominant N=328, subclonal N=968) and TNBC (right, clonally dominant N=121, subclonal N=190) breast cancers, for dominant and subclonal mutations. Signature contributions for clonal versus subclonal alterations were ascertained using deconstructSigs and compared using the Mann-Whitney U test. Signatures with significant difference in signature contribution and no overlap in interquartile range are identified with the p value.”

We have toned down conclusions around mutational signatures as requested. Additionally we supply orthogonal analysis using an additional mutational signature pipeline SigMA designed for smaller panels and providing further quantitative estimation of results and their robustness (Extended data Fig. 10).

Extended data Fig. 10: SigMA analysis of mutation signatures in clonally dominant and subclonal mutations in HR+HER2- and TNBC disease.

Signature analysis using SigMA⁶ for clonally dominant and subclonal mutations in HR+HER⁻ and TNBC breast cancer. NLS (non-negative least squares) exposure and Cosine similarity are two orthogonal methods which identify mutational signatures within sequencing data, whilst Likelihood describes the mutational signatures ascertained as present by the SigMA software. SigMA identifies APOBEC as strongly present in all HR+HER⁻ disease, whilst NLS exposure and Cosine similarity both identify signature 13 more strongly in subclonal disease than in clonally dominant disease. In TNBC, age-related signature 1 is more strongly identified by NLS exposure and Cosine similarity in subclonal disease than clonally dominant disease.

We additionally reanalysed signature contributions from deconstructSigs using a leave one out approach to assess the robustness of the calls (below). In HR+HER⁻ disease the contribution of signature 13 APOBEC-related signatures within subclonal

mutations remains stable when successive mutational signatures are removed. In TNBC disease the successive removal of signatures causes assignment of signatures that we would not expect to be present in breast cancer, likely confirming our conclusion that no single mutational process is important for generating genomic diversity in advanced TNBC.

We agree with the reviewer that clonal hematopoiesis is an important consideration. We have estimated the mutations arising from clonal hematopoiesis by looking at mutations occurring in genes on the panel commonly known to be involved in CHIP (*GNAS*, *JAK2*, *IDH1*, *IDH2* and *ATM*). These account for only 2.3% of the total mutations and we believe are therefore unlikely to have a large effect on the signature results. We have also compared the mutation distribution of mutations that could potentially arise from clonal hematopoiesis (n=68) to non-clonal hematopoiesis (n=2621, below). After considering the small numbers of potential clonal haematopoiesis mutations, we observe little difference in the profile of mutations which provides reassurance that any influence of clonal hematopoiesis on the mutation signature profiles identified is likely minimal.

REVIEWER #2 (REVIEWER COMMENTS TO THE AUTHOR):

Overview:

As the utility of circulating tumor DNA (ctDNA) continues to unfold in cancer, it is of certain value to reanalyze the genomic landscape of tumor types where the so called liquid biopsy approach might be adopted clinically. In this study, the authors take advantage of the samples and associated data collected (at baseline, pre-treatment) in the context of the plasmaMATCH trial, with metastatic breast cancer patients undergoing serial ctDNA testing in order to more precisely guide their treatment. This is an elegant work that “checks all the boxes” in this type of cancer genomics studies (description of the landscape, comparison to previously assembled datasets, study of clinical and pathological associations, etc), with the main novelties being the type of samples and the technology being used. The compendium of resulting observations, not only makes a nice companion paper to the trial main publication, but also offers a few potential new therapeutic approaches for advanced breast cancer, including some that might prevent acquired resistance to certain treatments. I don’t have major issues with the manuscript as it is. However, as the authors acknowledge, there are several important limitations to the study.

Comments:

1) It would be ideal that they discuss potential avenues to overcome these, like ways to deal with potential clonal hematopoiesis or qualifying samples for their approach by setting a threshold for ctDNA shedding in tumor samples. This could help with better assay development for future prospective studies that will continue to assess the accuracy of ctDNA analysis for routine practice and its potential to guide targeted therapy without requiring solid tissue testing (limited to dissect spatial heterogeneity and subclonal sampling).

We thank the reviewer for the suggestions. Clonal hematopoiesis can be effectively managed with either paired germline sequencing, or where not available, criteria (yet to be established in clinical research) can be used to remove potential CHIP mutations. Genes where CHIP may strongly contribute to mutation incidence (*GNAS*, *JAK2*, *IDH1*, *IDH2* and *ATM*), contribute to 2.3% of the data in the manuscript. Given that for the most part in breast cancer, genes affected by CHIP do not overlap with clinically important genes, and the estimated small number of mutations potentially attributable to CHIP, we do not consider this to be a major contributor to the data. We have amended the discussion on page 11 as follows to elaborate on clonal hematopoiesis:

“There are important limitations to our study and considerations to make when understanding the utility of ctDNA in clinical practice. Matched germline blood was not simultaneously sequenced in our study. A small subset of mutations we report in *TP53* and *KRAS* might have originated from clonal hematopoiesis as opposed to the cancer⁷. To confidently exclude clonal hematopoiesis future research could involve paired germline sequencing, or stringent criteria for variant identification in genes affected by clonal hematopoiesis.”

We agree with the reviewer that thresholds for ctDNA shedding would be helpful in clinical practice. ctDNA test results that are negative or low allele frequency mutations should be reviewed with caution as the potential for false negative results in these scenarios is high, driven by low purity of the sample. However DNA sequencing technology is rapidly evolving in this area, and there is high variability in the resolution of different sequencing techniques. It would therefore not be suitable to apply a static cutoff across the assays. Understanding which patients are *more likely* to have a false negative result would be informative, and our data presented here signals that those with fewer prior lines of treatment and soft tissue/nodal

disease as opposed to visceral disease have a lower maximum VAF (Fig. 3a). We have amended the discussion on page 11 as follows to highlight this issue.

“Consideration should also be given to the likelihood of false negative results, with some patient groups less likely to shed ctDNA (Fig. 3a), such as those with low burden nodal disease with fewer lines of therapy, where ctDNA may not fully characterise the mutations present.”

2) Speaking of tissue based sequencing, it would be appreciated if the authors provide some information about how the MSK-IMPACT cohort compares to the one in this trial regarding patient characteristics (for instance, lines of previous treatment).

Response: Unfortunately there is minimal available clinical data from the MSK-IMPACT dataset to allow meaningful comparison. We have compiled the following histological comparison with the available data, with the main significant difference being a higher proportion of patients with lobular breast cancer in the MSK-IMPACT dataset. We do not believe this will have significantly affected the comparison. This has been added to the manuscript as Extended data Tab. 1.

Extended data Tab. 1: Comparison of MSK-IMPACT cohort and plasmaMATCH cohort

Breast Cancer Subtype		plasmaMATCH			MSK-IMPACT			p value
		n cases	n	%	n cases	n	%	
HR+HER2-	Ductal	376	515	73.0	428	584	73.3	0.0003
	Lobular	63	515	12.2	107	584	18.3	
	Other/missing	76	515	14.8	49	584	8.4	
HR+HER2+	Ductal	35	46	76.1	56	75	74.7	0.28
	Lobular	2	46	4.3	9	75	12.0	
	Other/missing	9	46	19.6	10	75	13.3	
HR-HER2+	Ductal	20	26	76.9	35	43	81.4	0.62
	Lobular	1	26	3.8	3	43	7.0	
	Other/missing	5	26	19.2	5	43	11.6	
TNBC	Ductal	110	138	79.7	125	151	82.8	0.04

	Lobular	6	138	4.3	14	151	9.3	
	Other/missing	22	138	15.9	12	151	7.9	

p values from Chi Square test

3) One comment about the authors observations in TNBC. Given that the ctDNA genomics landscape in this subtype was similar to that of primary tissue sequencing, is it fair to conclude that ctDNA analysis is less informative in this subset of breast cancers? Is it because of the higher number of copy number alterations in TNBC? Would it be hard to replace tissue biopsy sequencing when it comes to copy number-based biomarkers assessment (as stated in one of the manuscripts), especially in this subtype?

Response: We agree that broadly there are similar profiles in advanced TNBC, compared to primary, likely reflecting the lack of targeted therapy in TNBC. As targeted, and immunotherapies, come to clinical use this may of course change. With regard to gene amplification, we demonstrated here that tissue defined *HER2* amplification is not sensitively identified in ctDNA using a targeted panel (Fig. 3d). We have adjusted the discussion on page 11 to acknowledge the limitation of copy number profile detection and potentially breast cancer subtype:

“Our study also emphasises that copy number detection is of limited sensitivity in plasma, and for tumor types where copy number events dominate tumor biology alternative approaches of genotyping are required⁸.”

4) Lastly, could the authors comment on the potential of more comprehensive sequencing approaches (WES) of ctDNA (as an alternative to approaches like the Guardant360 sequencing-based assay) to more accurately profile the mutation landscape of breast cancer patients in the future, including the identification of low frequency mutations, despite the lower read depth and the current lack of standardization of such approaches?

Response: We thank the author for this suggestion, and have amended the discussion on page 11 as follows (also as stated above) to elaborate on the relative benefits and drawbacks of different ctDNA sequencing methods.

“More comprehensive approaches to genotyping ctDNA such as whole exome sequencing could extend our observations. However, to detect subclonal mutations

such approaches will still require error correction, and such approaches are likely to be substantially expensive and likely beyond routine clinical application.”

REVIEWER #3 (REVIEWER COMMENTS TO THE AUTHOR):

Overview:

Authors should be commended on this excellent manuscript describing the plasmaMATCH cohort. Findings are interesting and highly relevant.

Comments

1) Comment should be made on patients that had no findings in their plasma. Who are these patients and what should be done with them?

Response: We thank the reviewer for this suggested additional analysis. Patients without alterations are statistically more likely to have had fewer lines of treatment than patients with alterations, which likely reflects the burden of disease. We have added a table of comparison to the supplementary data (Extended data Tab. 2), and amended the text on page 7 as follows:

“Using the rich clinical trial data available, we explored the clinical and pathological associations of ctDNA mutations, and the maximum variant allele frequency (mVAF) as a proxy of ctDNA purity (Fig. 3a). The number of lines of treatment was associated with increased number of SNVs/indels and mVAF (Fig. 3a), and soft tissue/nodal disease with lower mVAF (13.2 vs 8.0, $q=0.002$, Fig. 3a). Patients without a ctDNA alteration were significantly more likely to have had fewer lines of treatment ($p=0.015$, Extended data Tab. 2).”

Extended data Tab. 2: Comparison of Clinico-pathological characteristics of patients with and without ctDNA alterations identified

Clinical Characteristic		Patients with alterations n = 743		Patients without alterations n = 57		p value
		n	%	n	%	
Breast cancer	HR+HER2-	484	65.1	31	54.4	0.10
	HR+HER2+	40	5.4	6	10.5	

subtype	HR-HER2+	22	3.0	4	7.0	
	TNBC	130	17.5	8	14.0	
	Unknown	67	9.0	8	14.0	
Histology	Ductal	534	71.9	43	75.4	0.83
	Lobular	75	10.1	4	7.0	
	Other	43	5.8	4	7.0	
	Not known	91	12.2	6	10.5	
Disease burden	Visceral	586	78.9	41	71.9	0.41
	Soft tissue/nodal	131	17.6	12	21.1	
	Bone	10	1.3	1	1.8	
	Not known	16	2.2	3	5.3	
Number of lines prior treatment	0	67	9.0	10	17.5	0.015
	1-2	374	50.3	35	61.4	
	3-4	209	28.1	9	15.8	
	5+	93	12.5	3	5.3	

p values from Chi Square test

2) I would also consider bringing Extended Table 1 into the main manuscript as details about the metastatic patient population specifically lines of prior therapy as well as time since Dx to plasma sampling are important.

Response: We have moved Extended Table 1 to be in the main manuscript as requested.

3) To this end, can the authors comment on the Time categories and if more subclonal/clonal mutations are found with duration?

Response: We thank the reviewer for this suggested analysis, which we have undertaken. Our analysis indicates that there is no change in the proportion of alterations that are clonally dominant versus subclonal with time from primary breast cancer diagnosis (p=0.32). This is commented on on page 8 of the results:

“The proportion of dominant to subclonal mutations did not significantly alter with time from diagnosis of primary breast cancer (data not shown).”

References

1. Odegaard, J.I., *et al.* Validation of a plasma-based comprehensive cancer genotyping assay utilizing orthogonal tissue- and plasma-based methodologies. *Clin Cancer Res* (2018).
2. US Food and Drug Administration, F. SUMMARY OF SAFETY AND EFFECTIVENESS DATA (SSED). Vol. 2020 Guardant360® CDx FDA approval (2020).
3. Jacobs, M.T., *et al.* Use of Low-Frequency Driver Mutations Detected by Cell-Free Circulating Tumor DNA to Guide Targeted Therapy in Non-Small-Cell Lung Cancer: A Multicenter Case Series. *JCO precision oncology*, 1-10 (2018).
4. Aggarwal, C., *et al.* Clinical Implications of Plasma-Based Genotyping With the Delivery of Personalized Therapy in Metastatic Non-Small Cell Lung Cancer. *JAMA oncology* **5**, 173-180 (2019).
5. Turner, N.C., *et al.* Circulating tumour DNA analysis to direct therapy in advanced breast cancer (plasmaMATCH): a multicentre, multicohort, phase 2a, platform trial. *The Lancet Oncology*.
6. Gulhan, D.C., Lee, J.J.-K., Melloni, G.E.M., Cortés-Ciriano, I. & Park, P.J. Detecting the mutational signature of homologous recombination deficiency in clinical samples. *Nature Genetics* **51**, 912-919 (2019).
7. Razavi, P., *et al.* High-intensity sequencing reveals the sources of plasma circulating cell-free DNA variants. *Nature medicine* **25**, 1928-1937 (2019).

8. Comprehensive molecular portraits of human breast tumours. *Nature* **490**, 61-70 (2012).

REVIEWERS' COMMENTS

Reviewer #1 (Remarks to the Author):

The authors have addressed the most substantive comments, some minor points remain:

- the figures remain, disappointingly, unreadable at the resolution in the PDF file. Whether this is due to low resolution images or the conversion process I have no idea, but many of the legends and axes are unreadable and unlikely to be 5pt or greater at the size reproduced.

- the sensitivity/PPV analysis in response to original comment #3, above and below 1% VAF, should be incorporated into the manuscript.

- in response to comment #6, mutational signatures, the authors say in the discussion "In TNBC disease the successive removal of signatures causes assignment of signatures that we would not expect to be present in breast cancer, likely confirming our conclusion that no single mutational process is important for generating genomic diversity in advanced TNBC."

It would be more accurate to say that the assay used is not structured/does not have genomic coverage content to detect and distinguish mutational processes that are known to be important in TNBC (and in some ER+ cancers), such as homologous recombination repair deficiency, where genome scale copy number-structural aberration detection is required. The limited size of the panel also limits the ability in this regard to conclude that a single mutational process drives variation in ER+ cancers. The discussion should be amended to reflect these issues.

Reviewer #2 (Remarks to the Author):

After reading the revised manuscript about breast cancer ctDNA profiling in the context of the plasmaMATCH trial, I think that the authors have done a good job addressing this (and others) reviewer comments. I appreciate the authors making the effort to amend the text based on some of them and for the extended data added to the revision.

Response to Referees

Reviewer #1 (Remarks to the Author):

The authors have addressed the most substantive comments, some minor points remain:

1) the figures remain, disappointingly, unreadable at the resolution in the PDF file. Whether this is due to low resolution images or the conversion process I have no idea, but many of the legends and axes are unreadable and unlikely to be 5pt or greater at the size reproduced.

We apologise that the reviewer has had difficulty with suboptimal image resolution. We have reviewed all images to ensure that they comply with the formatting guidelines provided by Nature Communications. We will communicate with the graphics team to review the images to ensure comprehensibility.

2) the sensitivity/PPV analysis in response to original comment #3, above and below 1% VAF, should be incorporated into the manuscript.

This analysis has been added to the supplementary information as Supplementary Fig. 3, and commented in the Results section of the manuscript on page 5 as follows:

“We assessed the sensitivity of targeted sequencing to identify droplet digital PCR (ddPCR) mutation calls in targetable hotspots within *PIK3CA*, *HER2*, *AKT1* and *ESR1*. Within the 682 patients who underwent ctDNA testing with both technologies, the targeted sequencing demonstrated a high sensitivity of 90.9% in identifying mutations. For mutations with ddPCR allele frequency <1%, the targeted panel sensitivity was 80.9% (Supplementary Fig.3).”

3) in response to comment #6, mutational signatures, the authors say in the discussion "In TNBC disease the successive removal of signatures causes assignment of signatures that we would not expect to be present in breast cancer, likely confirming our conclusion that no single mutational process is important for generating genomic diversity in advanced TNBC."

It would be more accurate to say that the assay used is not structured/does not have genomic coverage content to detect and distinguish mutational processes that are known to be important in TNBC (and in some ER+ cancers), such as homologous recombination repair deficiency, where genome scale copy number-structural aberration detection is required. The limited size of the panel

also limits the ability in this regard to conclude that a single mutational process drives variation in ER+ cancers. The discussion should be amended to reflect these issues.

We agree with the reviewer that by the nature of the mutational signature analysis being derived from targeted sequencing data, we should be cautious in drawing conclusions from this analysis. We have altered the discussion in the text on page 11 to ensure that we do not imply that APOBEC mutagenesis is the sole driver of subclonal diversification, but rather a contributor:

“However, our finding of APOBEC mutations in HR+HER2- subclonal mutations concurs with tissue biopsy sequencing studies^{12,15}, and confirm that this process contributes to the subclonal diversity of HR+ advanced breast cancer. *PIK3CA* mutations vary in clonal dominance (Fig. 6a),”

We have also added further points in the discussion from page 11 to highlight the limitations of mutational signature analysis in targeted sequencing data, and also to stress the need for corroboration of these findings using broader sequencing data:

“Different mutational processes drive diversity in breast cancer subtypes. HR+ advanced breast cancer is characterised by subclonal mutations in part generated by APOBEC mutagenesis. We hypothesise that APOBEC is activated during prior endocrine therapy for advanced cancer, and that this may edit *PIK3CA* to generate frequent second hit novel mutations¹², resulting in hyperactivation of PI3K signalling⁴¹ and resistance to fulvestrant endocrine therapy (Fig. 6e). We performed a mutational signature analysis to differentiate drivers of clonal and subclonal disease. Relative to the broad sequencing approaches of whole-exome and whole-genome sequencing, targeted panels sequence selected areas of interest and as such cover less of the genome, which may limit mutational signature analysis on this data. However, our finding of APOBEC mutations in HR+HER2- subclonal mutations concurs with tissue biopsy sequencing studies^{12,15}, and confirm that this process contributes to the subclonal diversity of HR+ advanced breast cancer. *PIK3CA* mutations vary in clonal dominance (Fig. 6a), and future research will need to investigate whether this variation in subclonality effects response to PI3 kinase inhibitors in the clinic⁴². The poly-clonal nature of endocrine resistance likely substantially challenges attempts to treat endocrine resistant disease. Taken together, our findings emphasise the importance of investigating upfront combination

approaches to prevent endocrine resistance. Such approaches could possibly include APOBEC or PI3K pathway inhibitors.

In TNBC, we identified that subclonal diversity is associated with age related mutational signatures, suggesting a lack of specific processes driving subclonality, and potentially suggesting early diversification of metastatic TNBC. We note that patients with TNBC in this series were relatively infrequently treated with targeted therapies or immunotherapy, and it is possible that in the future specific mutational processes selected by targeted therapies will shape subclonality in TNBC. Given the limitations of mutational signature analysis undertaken in targeted sequencing data, these findings require corroboration with whole-exome or whole genome data.”

Reviewer #2 (Remarks to the Author):

After reading the revised manuscript about breast cancer ctDNA profiling in the context of the plasmaMATCH trial, I think that the authors have done a good job addressing this (and others) reviewer comments. I appreciate the authors making the effort to amend the text based on some of them and for the extended data added to the revision.